# Plants or bacteria? 130 years of mixed imprints in Lake Baldegg sediments (Switzerland), as revealed by compound-specific isotope analysis (CSIA) and biomarker analysis

Marlène Lavrieux[1*], Axel Birkholz[1*], Katrin Meusburger[1,2], Guido L.B. Wiesenberg[3], Adrian Gilli[4], Christian Stamm[5], Christine Alewell[1]

[1]Environmental Geosciences, Department Environmental Sciences, University of Basel, Basel, Switzerland
[2]Swiss Federal Institute for Forest, Snow and Landscape Research, Birmensdorf, Switzerland
[3]University of Zurich, Department of Geography, Zurich, Switzerland
[4]Geological Institute, ETH Zurich, Zurich, Switzerland
[5]Eawag, Swiss Federal Institute of Aquatic Science and Technology, Dübendorf, Switzerland
* These authors contributed equally to this work.

*Correspondence to*: Axel Birkholz (axel.birkholz@unibas.ch)

**Abstract.** Soil erosion and associated sediment transfer are among the major causes of aquatic ecosystem and surface water quality impairment. Through land-use and agricultural practices, human activities modify the soil erosive risk and the catchment connectivity, becoming a key factor of sediment dynamics. Hence, restoration and management plans of water bodies can only be efficient if the sediment sources and the proportion attributable to different land-uses are identified. To this aim, we applied two approaches, namely compound-specific isotope analysis (CSIA) of long-chain fatty acids (FA) and triterpenoid biomarker analysis, to the eutrophic Lake Baldegg and its agriculturally used catchment (Switzerland). Soils reflecting the five main land-uses of the catchment (arable lands, temporary and permanent grasslands, mixed forests, orchards) were subjected to CSIA. The compound-specific stable isotope $\delta^{13}C$ signatures clearly discriminate between potential grasslands (permanent and temporary) and forest sources. Signatures of agricultural land and orchards fall in-between. The soil signal was compared to the isotopic signature of a lake sediment sequence covering ca. 130 years (before 1885 to 2009). The recent lake samples (1940 – 2009, with the exception of 1964-1972) fall into the soil isotopic signature polygon and indicate an important contribution of the forests, which might be explained by (1) the location of the forests on steep slopes, resulting in a higher connectivity of the forests to the lake, and/or (2) potential direct inputs of trees and shrubs growing along the rivers feeding the lake and around the lake. However, the lake sediment samples older than 1940 lie outside the source soils polygon, as a result of FA contribution from a not yet identified source, most likely produced by an in-situ aquatic source, either algae, bacteria or other microorganisms or an ex-site historic source from wetland soils and plants (e.g. Sphagnum spec.). Despite the overprint of the yet unknown source on the historic isotopic signal of the lake sediments, land-use and catchment history are clearly reflected in the CSIA results, with isotopic shifts being synchronous with changes in catchment, land-use and eutrophication history. The investigated highly specific biomarkers were not detected in the lake sediment even though they were present in the soils. However, two trimethyltetrahydrochrysenes (TTHCs), natural diagenetic products of pentacyclic triterpenoids, were found in the lake sediments. Their origin is attributed to the in-situ microbial degradation of some of the triterpenoids. While the need to apportion sediment sources is especially crucial in eutrophic systems, our study stresses the importance of using caution with CSIA and triterpenoid biomarkers in such environments, where the active metabolism of bacteria might mask the original terrestrial isotopic signals.

## 1 Introduction

While it is known that pollutant inputs have a severe impact on aquatic ecosystems, especially in agriculturally used catchments (Malaj et al., 2014; Allan, 2004; Liess et al. 2001), the influence of sediment input and sediment dynamics on biological quality

and recovery of rivers remains highly uncertain (Scheurer et al., 2009; Matthaei et al., 2010). Sediment loads to freshwaters are increasing worldwide, often being related to anthropogenic activities (Scheurer et al. 2009). Sediment pollution has been identified as one of the most relevant pressures to water bodies (Borja et al., 2006), and sediments are among the top ten causes of biological impairment in freshwater ecosystems (US EPA, 2009). Land-uses and agricultural practices modify the soils

erosive risk and the catchments sedimentary connectivity, becoming a key factor of sediment dynamics and aquatic ecosystems health. Restoration and management plans of water bodies can only be efficient if the sediment sources and their respective contributions, i.e. the proportion attributable to different land uses are identified (Wasson et al. 2010; Sundermann et al. 2013).

The compound-specific isotope analysis (CSIA) technique, based on the compound specific stable isotope signatures of inherent organic biomarkers in the soil, was developed and applied to discriminate and apportion the source soil contribution

from different land-uses (Gibbs, 2008; Blake et al., 2012; Hancock and Revill, 2013; Alewell et al., 2016). The FAs being transferable from plants to soils, stable, persistent in soils, mobile with sediments during flow events and easily isolatable from the other compounds in lipid mixtures, they are especially well suited for CSIA (Reiffarth et al., 2016). While FAs assemblages are not variable enough among plant species to discriminate them, their $\delta^{13}C$ signature differs among groups of plant species (Tolosa et al., 2013). The $\delta^{13}C$ signature of biomarkers is assumed to be more preserved than their concentration during

degradation and transport processes (e.g. Marseille et al., 1999; Gibbs, 2008), allowing to discriminate sources in various studies in lake sediment and catchment studies (e.g. Galy et al., 2011; Fang et al., 2014), even dominated by C3 vegetation only (Alewell et al., 2016).

In addition to the CSIA, attention was given to some cyclic compounds as specific tracers for source identification. A large part of the cyclic compounds is synthesized by more restricted plant groups than linear alkyl lipids. Among the cyclic

compounds, some triterpenes were validated as family- or even species-specific (e.g. some triterpenyl acetates for Asteraceae, some sesqui-, di- and triterpenoids for conifers, methoxyserratenes for Pinaceae; Lavrieux et al., 2011; Otto and Wilde, 2001; Le Milbeau et al., 2013; respectively). Mostly developed and successfully used for paleo-environmental studies (e.g. Jacob et al., 2008; Lavrieux, 2011; Guillemot et al., 2017), the high potential of these highly specific biomarkers (HSB) for tracking sediment sources and evaluating the soil vulnerability remains under-exploited.

The need to precisely identify sediment sources is especially important in eutrophic systems to enable efficient and targeted restoration measures. For this reason, we chose to use a mixed CSIA and HSB approach to the Lake Baldegg catchment (Central Switzerland). The eutrophic Lake Baldegg is a typical but also extreme example of a European freshwater body, as it suffered substantially from nutrient input (mainly phosphorus, P) during the second half of the 20[th] century. Studies have been carried out on the P source attribution into the lake but the origin of sediments remains unclear. While the eutrophication

history of the Lake Baldegg has extensively been studied (e.g. Niessen and Sturm, 1982; Lotter et al., 1997; Lotter, 1998; Teranes and Bernasconi, 2005), an in-depth confrontation of the lake evolution with the recent history of the catchment (including land-use and agricultural practices changes) has not yet been performed.

Our project aimed at filling these gaps. In this paper, the soil isotopic signatures of FAs characterizing the main land-uses of Lake Baldegg catchment are quantified and confronted to the evolution of the CSIA imprint of a 130-yrs long lake sediment

sequence. This study is, to our knowledge, the first sediment fingerprinting CSIA concerning a lake sediment core covering more than a century.

## 2 Study site

Lake Baldegg (N47°12' 0", E8°15'40"; 463 m a.s.l.) is a eutrophic lake of glacial origin located on the central Swiss Plateau (Fig. 1). It has a maximum depth of 66 m, a surface area of 5.2 km$^2$ and a water volume of 0.173 km$^3$. The lake is fed by 15 streams and has a mean residence time of 4.3 years (Wehrli et al., 1997). The outflow is located at its northern end. Its North-South catchment, having an area of 67.8 km$^2$, has hillslopes of 700-800 m a.s.l. elevation. The catchment is today intensively used for agriculture: 77% is used as agricultural land, 12% as forest (mostly on the slopes), 5% as urbanized areas (Wehrli et al., 1997). In 2015, one third of the agricultural land was devoted to permanent grassland, 40% to cereals and arable lands (including 10% of maize), 24% to temporary grasslands, while fruit production (small trees, mainly apples and pears) covered ca. 1% of the agricultural land (Federal Statistical Office, 2015). Intensive chicken farming and pig breeding are other important farming activities.

Previous studies have provided extensive information about the lake eutrophication history (e.g. Lotter et al., 1997, 1998; Wehrli et al., 1997). Briefly, this eutrophication, starting in 1885, translated into annually laminated (varved) sediments in a context of constant anoxic lake bottom until the 1980s (anoxia below 60 m depth between 1885-1940; below 40 m between 1940-1970; below 10 m between 1970-1982; Niessen and Sturm, 1987; Lotter et al., 1997). Along the 20$^{th}$ century, a severe increase in phosphorus loads stemming from the intensification of land-use, population and industrial activities, supported an increase in the eutrophication. The almost exponentially increasing phosphorus concentration in the lake water (up to > 500 µg.l$^{-1}$; Wehrli et al., 1997), leading to hypereutrophic conditions with dramatic fish kills and algal blooms, was curbed after the introduction of wastewater treatment plants and several restoration efforts. The introduction of an artificial oxygenation system into the lake water column in 1982 (Stadelmann et al., 2002) lead to the disappearance of the varves from 1995. Despite the accompanying strong decrease of P concentrations in the lake to below 30 µg.l$^{-1}$ as the result of lake external and internal measures, the lake has not yet fully recovered from eutrophication (Müller et al, 2014).

## 3 Materials and methods

### 3.1 Connectivity index

With the purpose to sample the source soils most likely contributing to the recovered lake sediment, a connectivity index model and a connectivity map were built. Connectivity patterns in the catchment were identified using a modified sediment connectivity index (IC) based on the approach by Borselli et al. (2008) and modified by Cavalli et al. (2013) (Fig. 2). This index, calculating surface roughness from a high-resolution digital elevation model (2m resolution, swissALTI3D), indicates the degree of linkage controlling sediment fluxes throughout landscape, and, in particular, between sediment sources and downstream areas and finally the freshwater system.

### 3.2 Sampling

#### 3.2.1 Soils

Soil sampling locations were chosen according to the abovementioned connectivity model approach, the land-use map (Fig. 2) and aerial photographs. The focus of this study was set on areas with high connectivity. Soil samples representing each main land-use type (arable lands, permanent grasslands, temporary grasslands, mixed forests, orchards) were taken. Five sites were selected for orchards and forests, four sites for arable lands and temporary grasslands, and three sites for permanent grasslands. Within each site, four soil cores were sampled and mixed into a composite sample. For all forest sites the humus layer was removed prior to sampling. At four of five investigated forest sites no Oa layer was present (only partly degraded material (Oe) and intact plant material (Oi)). Only at one site (Norwegian spruce and *Thuidium tamariscinum* moss) an Oa

layer had built up. For the orchards, samples were taken at the base of the trees, where no herbaceous vegetation was growing. Distinction between temporary and permanent grasslands was made from the vegetation diversity observed on the field, and the presence of a tilled horizon was checked with a Pürckhauer auger system. The 5 uppermost centimeters of the soil were sampled with a 5-cm high cylindrical steel ring (98.2 $cm^3$) and stored in aluminum foil in the fridge until drying.

### 3.2.2 Lake sediment core

We subsampled in January 2016 a sediment core (Ba-09-03) retrieved in autumn 2009 in the deepest part of Lake Baldegg, which was stored in a refrigerated storage room since then. The varved sediment allows dating of the cores at a seasonal resolution back to 1885 CE. Detailed retrieving and sediment core information, as well as the age-depth model, is documented in van Raden (2012) and Kind (2012). The upper 45 cm of the core, covering the last 130 years, were sampled in 3 years slices. The 9-mm-thick turbidite of 1956 CE was sampled apart. Every second sample between 1885 and 2009 CE, as well as one sample older than 1885 CE, i.e. before eutrophication start, were further analyzed. The oldest sample was dated to ca. 1870 CE by extrapolating the sedimentation rate of the well-dated last 19[th] century varved part.

### 3.3 Sample preparation

After freeze-drying (lake sediments) or oven-drying (soils; 40°C, 72 hours), the sediment samples were carefully crushed with a pestle and mortar. Soils were dry sieved at 2 mm, which was not necessary for the fine-grained lake sediment. With great care the macroscopic elements (vegetal remains, stones) were hand-picked from all the samples. 2-4 g of samples (soils and lake sediments) were processed for the lipid extraction, using a mixture of $CH_2Cl_2$:MeOH (9:1 v/v) in an Accelerated Solvent Extractor (Dionex ASE 200 for the lake sediments, Dionex ASE 350 for the soils). Lipid extracts were subsequently separated into neutral, acidic and polar fractions using solid-phase extraction on aminopropyl-bonded silica as described in Jacob et al. (2005).

### 3.3.1 Fatty acid preparation for CSIA

The acidic fraction (including the free fatty acids) was methylated at 60°C for 1h using 1 mL of 12–14% $BF_3$ in MeOH. Fatty acid methyl esters (FAMEs) were extracted from the solution by agitating four times with ca. 2 mL hexane in the presence of 1 mL of 0.1 M KCl. The final extract was stored in the freezer until analysis.

The purity of the extract and the concentration of the FAMEs were checked using a Trace Ultra gas chromatograph (GC) with a flame ionization detector (FID; Thermo Scientific, Walthalm, MA 02451, USA) as described in Alewell et al. (2016). Lake sediments FAMEs stable carbon isotopic composition was measured as described in Alewell et al. (2016) using a Trace Ultra GC, coupled via combustion interface GC Isolink and Conflo IV with a Delta V Advantage isotope ratio mass spectrometer (Thermo Scientific). Soils FAMEs stable carbon isotopic composition was measured using a Trace 1310 GC instrument interfaced on-line via a GC-Isolink II to a Conflo IV and Delta V Plus isotope ratio mass spectrometer (Thermo Fisher Scientific). A DB 5ms column (J & W DB-5MS, 50 m × 0.2 mm i.d., 0.33 µm film thickness) was used. The GC temperature program was 70 °C (held 4 min) to 150°C at 20°C/min and afterwards to 320 °C (held 40 min) at 5 °C/min. He was used as carrier gas at a constant 1 ml/min. $CO_2$ of known $\delta^{13}C$ composition was automatically introduced via Conflo IV into the isotopic ratio mass spectrometer in a series of 5 pulses at the beginning and 4 pulses the end of each analysis, respectively, and used as reference gas during every measurement. The comparability of soils and lake sediment results was ensured by triplicate measurements of 3 lake samples realized on both instruments. Each sample was measured at least 3 times. Carbon stable isotope ratios were reported in delta notation, per mil deviation from Vienna Pee Dee Belemnite (VPDB). The instruments performance was routinely checked with an external isotopically characterized fatty acids mixture (F8-3) obtained from Arndt Schimmelmann (see http://pages.iu.edu/~aschimme/hc.html), to which a mixture of isotopically characterized C24:0, C26:0,

C28:0 and C30:0 FAMEs was added. Performance was controlled with a C19:0 FA internal standard. The reported $\delta^{13}$C values were corrected for the additional carbon atom introduced during methylation. Mean values of at least triplicate measurements, as well as their corresponding standard deviation, were calculated. The analytical uncertainty is lower than ±0.5 ‰. Only long-chain FAs ≥ C24:0 were investigated. These are characteristic for the higher plant input into the soil (Eglinton and Eglinton, 2008). Short- or mid-chain FAs can also be produced by bacteria or aquatic plants and would bias our approach to trace back the terrestrial input into the lake.

### 3.3.2 Triterpenoids

The neutral fraction (including the cyclic biomarkers considered in this study) was further separated into aliphatics, aromatics, ethers and esters, ketones and acetates, and alcohols by flash chromatography on a Pasteur pipette filled with activated silica (24 h at 120 °C, then deactivated with 5% $H_2O$) and using a sequence of solvents of increasing polarity. The alcohol fraction was silylated before injection by reaction with N,O-bis-(trimethylsilyl)trifluoroacetamide (BSTFA) containing 1% trimethylchlorosilane and pyridine for approximately 1 h at 60°C. 5α-cholestane, which was used as an internal standard, was added to all fractions, prior to analysis by gas chromatography-mass spectrometry (GC-MS) with a Trace GC Ultra coupled to a DSQII mass spectrometer (Thermo Fisher Scientific). The GC instrument was fitted with a Restek Rxi-5ms column (60m x 0,25mm i.d., 0.25µm film thickness). Samples were injected in splitless mode, with the injector temperature set at 300 °C. He was the carrier gas at a constant flow of 1.2 ml.min$^{-1}$. The GC temperature program was 50 °C (held 2 min) to 140 °C (held 1 min) at 10 °C/min, then to 300°C (held 63 min) at 4°C/min. The transfer line to MS detector was operated at 260°C. The mass spectrometer was operated in the electron ionization (EI) mode at 70eV and scanned from m/z 40 to 1000. Component identification was based on comparison with literature data.

## 4 Results and discussion

### 4.1 CSIA of potential source soils

Among the FAs detected in soils (C17:0 to C32:0), only the longer chains, i.e. longer than C24:0, were further considered for this study to limit errors due to aquatic organisms contribution (Alewell et al., 2016). Though present in soils, C30:0 and C32:0 were not further considered here as their too low concentration (C30:0) or absence (C32:0) in the lake sediments hampers their use for fingerprinting. Fig. 3 displays the CSIA isoplots for the C24:0 vs. C26:0, C26:0 vs. C28:0, and C24:0 vs. C28:0. Data are provided in Table S1. The C26:0 vs. C28:0 show the best discrimination between the different land-use types.

All the samples align along a line, which ends are the isotopic signals of the grasslands and the forests soils. Halfway between them, orchard signature probably holds a mix between the inputs of the fruit trees, which signature might be supposed to be comparable to forest trees, and of the underlying grass. One orchard sample plots within the forest pool. Being covered of the same tree species as the other orchards (apple trees), and the age of the orchard having no influence on the measured imprint (Table S3), it is most probable that the corresponding sample was taken nearer from the trees than the other ones. CSIA signatures of arable lands plot near the orchards. The good separation between grasslands and forest pools confirm the results published on the Enziwigger catchment (ca. 30 km West of Lake Baldegg; Alewell et al., 2016), but our results show a better distinction between arable lands and grasslands – which could not be separated in this previous study. This can be either due to the greater surface covered with maize in Lake Baldegg catchment (ca. 10% of agricultural land in 2015; Federal Statistical Office) compared to Enziwigger catchment, where the low maize production does not produce any detectable effect on the stable isotope signature of soils (Schindler Wildhaber et al., 2012) or to more frequent rotation of grasslands and arable crops

in the Enziwigger study. As we cannot exclude for temporary grasslands to be part of crop rotations including cereals, we expected temporary grasslands to plot near arable lands at Lake Baldegg. But CSIA signatures cannot distinguish between non-permanent and permanent grasslands. As turnover times of one to several decades were reported for lipid fractions in croplands, permanent grasslands and forests (Wiesenberg et al., 2004; Wiesenberg et al., 2008; Griepentrog et al., 2015, respectively), the rapid loss of an arable land imprint after rotation to grassland seems unlikely. Most probably, the corresponding non-permanent grasslands are, even though regularly ploughed and the vegetation regularly re-sowed, used mostly as grasslands for many years, resulting in an imprint comparable to the permanent grasslands one. Further inquiries with local farmers confirmed, that most temporary grasslands were just plowed and reseeded to control for homogenous and highly productive species distribution.

## 4.2 CSIA of lake sediments

Considering the very low concentration of the C30:0 FA in the lake sediment, only the C24:0, C26:0 and C28:0 homologues were considered here to avoid any biases due to concentration effect (Fig. 4). Data are provided in Table S2. The isotopic signature of the samples older than 1940 and from 1964 - 1972 fall out of the source soils mixing polygon, making the use of a mixing model to quantify the contribution of different land-uses to sediment inputs impossible. This mismatch between soil and most of the lake sediment signals indicates that we did not catch all contributing sources to the lake sediment FA isotopic signal.

4.2.1 The likelihood of missing an additional terrestrial source to the isotopic FA signal of lake sediments

Land-use and land-use change is exceptionally well documented in this Swiss catchment. Vegetation composition did not dramatically change during the last century and to our knowledge there are no plausible additional land-use types as soil sources to the lake sediments we might have missed over the last decades. Any input from sewage sludge or from pig faeces originating from the intensive farming attested since the mid-1960's around the lake can be excluded, even before the introduction of wastewater treatment plants in the late 1960's, since both are not known as sources of long-chain saturated FAs (Cummings, 1981; Jørgensen et al., 1993; Jardé et al., 2005; Réveillé et al., 2003, respectively).

The input of organics from humus material and mixing into the lake sediments might be discussed as a potential additional source. However, great care was paid to remove any macroscopic organic material from the sieved soil and lake sediment samples. Even if we missed some highly decomposed Oa material, a study about fractionation processes of FAs in the humus layer of forest soils at Baldegg Lake catchment recorded no or only slight changes in the isotopic signal from Oa to Ah horizon (unpublished data). For our site with the Oa horizon, C28:0 and C30:0 FA were only slightly depleted by 0.2 - 0.3‰ compared to the Ah horizon. C24:0 and C26:0 were depleted by 0.8 and 1‰ respectively. But these humus $\delta^{13}C$ values are -33.8‰ for C26:0 FA and -34.6‰ for C28:0 FA and can thus not explain shifts of C26:0/C28:0 to values more negative than -36‰ (compare Fig. 4). Further, these humus $\delta^{13}C$ values still lie in the isotopic range of the five analyzed forest locations (Fig. 4) and would not be separable as a discrete source. Also, as today's isotopic signals of lake sediment samples plot within the polygon of the source soil signals, we rather expect a source or process different from today's conditions being the cause for the deviation of isotopic signals.

Historical research (Kopp, 1962) has revealed that the lake level was lowered by 30-40 cm at the beginning of the 19[th] century. This lake level change has changed the hydrology of riparian zones and wetlands, which have drained into the lake and were drained by the farmers to use the fertile riparian area. As such, organic material from wetland soils (e.g., fens, riparian zones) might have been leached and eroded due to the change in hydrological regime and/or drainage of sites due to adapted land use.

Furthermore, established reedlands, with *phragmites australis*, next to the main inflow at the southern end of the catchment have been cut starting in 1944 and in 1955 they completely drained this reedland. Today this area is a small lake with small surrounding wetlands, still containing *phragmites australis* (Rezbanyai, 1981). As such, another possible explanation for the negative values of the FAs C24:0 and C26:0 could be a larger contribution of wetland organic matter derived from e.g.

*phragmites australis* or *sphagnum* species, to the eroded sediments. Especially *sphagnum* species comprise high concentrations of C24:0 and C26:0 FAs (Baas et al., 2000, Pancost et al., 2002). Photosynthesis of *Phragmites* or mosses like *sphagnum* in the riparian zone or in peats respectively with $CO_2$ derived from oxidized methane could be an optional source for depleted long-chain fatty acids (cf. Alewell et al. 2011, $d^{13}C$ depletion of mosses, induced by photosynthesis with methane derived $CO_2$, effected the bulk carbon $d^{13}C$ in Scottish bog). However, $\delta^{13}C$ values of long-chain FAs from a Scottish peat core were not

depleted and range between -29.5‰ and -32.8‰ (Ficken et al., 1998) and would not be an adequate explanation for our missing source. Nevertheless, a depleted $\delta^{13}C$ of C24:0 and C26:0 FAs from wetlands or peat could explain the deviation for most of our lake sediment samples. Even more so, if we consider the relatively low C28:0 FA concentration in Sphagnum (Baas et al., 2000, Pancost et al., 2002) compared to C24:0 and C26:0 FAs. The larger proportions of C24:0 and C26:0 FA would clearly dominate over the C28:0 FA signal and could explain the stronger shift of the lake sediments to more depleted C24:0 and

C26:0 FA $\delta^{13}C$ values compared to a C28:0 FA $\delta^{13}C$ signal, dominated by the other land-use types. This hypothesis can not be tested in retrospect, as these reedlands/wetlands disappeared 50 – 75 years ago, but seems one possible explanation for the isotopic depletion of the FAs before 1940, especially if we consider the high concentration of C24:0 in sphagnum (cf. Baas et al., 2000, Pancost et al., 2002).

4.2.2 The likelihood of missing an in-situ source to the isotopic FA signal of lake sediments

One potential missed source might be the influence of complete organisms or residues of e.g. chironomid larvae on the $\delta^{13}C$ signal of the lake sediment. Fatty acids produced by these larvae might be depleted in $\delta^{13}C$ (Makhutova et al., 2017) and could act as a potential depleted source. However, sample preparation was done in paying attention to the possible occurrence of chironomid larvae, and we can thus exclude the presence of these organisms in relevant amounts. Moreover, no literature was

found stating the production of FAs longer than C22:0 by these larvae.

Other sources for long-chain FAs might be lacustrine macrophytes and microbial organisms (e.g. Volkmann et al., 1988; Volkmann et al., 1998; Bovee and Pearson, 2014; Schouten et al., 1998; van Bree et al., 2018), but no reference to the production of long-chain FAs by organisms known to live in the Lake Baldegg could be found: the algae responsible for the blooms (toxic blue algae *Aphanizomenon* and *Anabaena* during the 1960's, green algae *Pediastrum* especially between 1965-

1970; Stadelmann et al, 2002; van der Knaap et al., 2000) are indeed not reputed producing long-chain saturated FAs (Gugger et al., 2002; Caudales and Wells, 1992; Parker et al., 1967; Blokker et al, 1998). Very recently van Bree et al. (2018) were suggesting production of long-chain FAs, mainly C28:0, in the water column by algae or bacteria, while studying suspended particulate matter (SPM) from Lake Chala (Kenya/Tanzania). They draw their conclusions from a very strong seasonal variability in the SPM, different timing of the maximum concentrations of long-chain *n*-alkanes and long-chain fatty acids,

and very negative $\delta^{13}C$ values down to -46.3‰ for C28:0 FA and -41.9‰ for C26:0 FA. One possible explanation for the very depleted $\delta^{13}C$ values of the FAs might be the $CO_2$ uptake of the lake surface water. During times of under saturation with $CO_2$ and high pH values (8.3-9) atmospheric $CO_2$ is reacting with $OH^-$ to $HCO_3^-$ (Hydroxilation of $CO_2$). This reaction results in a strong carbon isotopic fractionation of -12‰. This highly depleted $HCO_3^-$ can heavily influence the isotopic signal of FAs

produced by aquatic organisms (van Bree et al., 2018 and references therein; Terranes et al., 1999 and reference therein, describing the same process for depleted $\delta^{18}O$ in Lake Baldegg).

As we have generally conditions of $CO_2$ undersaturation in Lake Baldegg beginning at the end of April when stratification is starting and epilimnic primary production is increasing (Müller et al., 2016), the above described incorporation of atmospheric
$CO_2$ into FAs might also serve as a possible explanation for the depleted values of C24:0 and C26:0. Also pH values during that time of the year are above 8.3 and reach 8.6 during June (Terranes et al., 1999) and hydroxilation of $CO_2$ will be a dominant process.

We noted that the shorter the homologue is, the more deviated from the soils polygon its isotopic values are: while the soil values have a range of ±3.5‰ for C24:0, ±2.6‰ for C26:0, and ±3.2‰ for C28:0, the lake values have a range of ±9, ±5 and
±5‰, respectively. For long-chain FAs of close chain-length, the $\delta^{13}C$ values are generally comprised in a range of a few permil because of their common biosynthesis pathway (Hayes, 1993; Wiesenberg et al., 2004). However, we observe considerable differences between isotopic signatures of the long-chain FAs. Both, the greater variation of the CSIA values in the lake compared to the soils as well as the discrepancies of up to 7 permil between C24:0 and C28:0, as observed here in lake sediments, suggest that an aquatic process might have masked the terrestrial isotopic signatures. Maybe similar to Lake
Chala (van Bree et al., 2018), in Lake Baldegg the FAs C24:0 and C26:0 are primarily produced by an algal or bacterial source whereas C28:0 still reflects the signal of the terrestrial vegetation. However, we do not see any increase in concentration of C24:0 or C26:0 FA in the lake sediments until 1940 compared to today's soils (compare Fig. S1 and S2 in the supplementary material). We would expect higher FA concentrations in lake sediments compared to source soils with a significant enhanced in-situ production in the lake water column. However, we do not know the historic FA content of source soils.

The depleted $\delta^{13}C$ values might also be linked to bacterially assimilated carbon, associated to anoxic conditions in the water column, sediments or wetland soils (Summons et al., 1994; Teranes and Bernasconi, 2005). Biogenic methane carbon typically shows $\delta^{13}C$ values of -50 to -70‰ (Whiticar, 1999), leading to a very depleted methanotrophic bacterial biomass (e.g. Summons et al., 1994; Lehmann et al, 2004). The influence of the methanotrophic bacterial communities in the Lake Baldegg was already underlined by the study of Teranes and Bernasconi (2005). A $\delta^{13}C$ value of -70+- 15‰ for methanotrophic bacteria
using biogenic methane can be assumed (Lehmann et al., 2004 and references therein). In this case only little bacterial biomass would be needed to cause depletion effects like we observe in the Lake Baldegg sediments. As well, $\delta^{13}C$ depleted $CO_2$ produced by methane-oxidizing bacteria (MOB) would result in the depleted long-chain FAs. Algae or Cyanobacteria could take up this depleted $CO_2$ and produce on their part depleted FAs (Naeher et al., 2014). But the presence of these long-chain $n$-fatty acids produced in the lake seemed so far to be unlikely here, since to our knowledge, reports about the production of
long-chain $n$-fatty acids by bacteria or algae are rare and constrained to extreme environments (e.g. Antarctic Ace Lake, Volkman et al., 1988; Volkman et al., 1998; van Bree et al., 2018). Gong and Hollander (1997) were suspecting marine bacteria contributing depleted long-chain FAs to the formerly assumed terrestrial long-chain FA pool in marine sediments. Also Feakins et al. (2007) described the in-situ production of long-chain FAs in a lacustrine environment by algae or bacteria as very likely. In their study not the depletion of the FAs was leading to this conclusion but the ratios between $n$-alkanes and FAs.

In a recent publication, Petrisic et al, (2017) were reporting $\delta^{13}C$ values of -40.0 - -43.3‰ for C26:0 FA from surface sediments from Lake Bled, Slovenia. They suspected either methanotroph bacteria or cyano bacteria as the producers of C26:0 FA. Pertrisics findings from Lake Bled are giving strong evidence that bacteria can play an important role for in-situ production of long chain FAs also in lakes beyond extreme environments. This leaves the open question why C24:0 FA in the Lake Bled sediments is not, or only slightly depleted with $\delta^{13}C$ values ranging from -36.3 - -37.7‰. Unfortunately, no data for C28:0 FA

from Lake Bled is available (Petrisic et al., 2017). As explained above, the influence of the unknown source in Lake Baldegg sediments is increasing from C26:0 to C24:0 FA. However, Neunlist et al. (2002) presented isotopic data for C24:0 – C28:0 FA from Lake Bled and Lake Baldegg sediment samples. Interestingly, in the two more recent sediment samples from Lake Bled, 1993-1996 and 1984-1990, no depletion of the $\delta^{13}C$ values was observed, whereas the two older sample from 1967-1976 and 1943-1956 show a clear isotopic depletion. The Authors were suggesting a continuous change in the isotopic composition of the carbon source of the producing organisms. For Lake Baldegg, four sediment samples from different depths were analyzed, but in contrast to our study, no significant depletion of the FA $\delta^{13}C$ values was observed. However, none of the investigated samples originated from a time where we observed the strong isotopic depletion of C26:0 and C24:0 FA in Lake Baldegg. Neunlist et al. (2002) concluded, that there were constant stable sources for the linear compounds and a higher plant origin very likely.

Taking all this into account the likelihood of an in-situ production of long chain FAs by aquatic organisms like algae or bacteria is given. The most likely were algal (phytoplankton) production caused by uptake of depleted $HCO_3^-$ due to undersaturation of $CO_2$ in the surface water, high pH values and associated hydroxylation of $CO_2$ (van Bree et al., 2018) or production by methanotroph bacteria and/or by algae or Cyanobacteria which used $CO_2$ depleted in $\delta^{13}C$, formerly produced by MOB (Naeher et al., 2014; Petrisic et al., 2017).

4.2.3 The necessity of "Suess effect" correction for terrestrial lipids in lake sediments

The $\delta^{13}C$ value of atmospheric $CO_2$ has decreased by approximately 1.5‰ since the beginning of the industrial era in response to fossil fuel combustion (atmospheric $\delta^{13}C$ = -6.5‰ in the preindustrial era vs. -8‰ today; Rubino et al., 2013; Keeling et al., 2005, respectively). Therefore, we would expect older lake sediment samples to be relatively enriched (less depleted) in $\delta^{13}C$ compared to our todays source soils or sediments. As such, the Suess effect can not explain our deviation of the isotopic signals between source soils polygon and lake sediments. However, it is widely recognized as necessary to correct for the Suess effect in autochthonous organic matter (Verburgh 2007). We thus want to discuss in the following the application of this correction for terrestrial derived organic matter in lake sediments (in our study long-chain FAs as biomarker for higher plants (Eglinton and Eglinton, 2008)).

Long-term experiments have shown that because a depletion due to the Suess effect is well recorded in plants (e.g. Zhao et al., 2001), this effect should also be recorded in soils, and consequently also in organic terrestrial markers archived in lake sediments, such as FAs. However, the Suess effect can only account for a maximum decrease of ca. 1.5‰ in the atmosphere, or taking the dataset of Zhao et al. (2001) for straw from 1845 until 1997 into account, for a maximum of 2.5‰. And it might only have an influence in the case of a soil having a fast turnover time of the overall soil organic matter of one to a few years (Garten et al., 2000). Longer turnover times imply necessarily a time lag in the recording of the Suess effect in soils accompanied with a strong dampening of the incoming isotopic signal .

To estimate the Suess effect on our soils and sediments we did a correction based on the atmospheric $CO_2$ curve of Feng (1998) We applied the following equation (1) to calculate the Suess effect induced changes from year to year with 1840 as the starting point.

$$\delta^{13}C_{Soil\,(t+1)} = (1-1/R)\,^{13}C_{Soil\,(t)} + 1/R\,(\delta^{13}C_{Soil\,(t0)} + (\delta^{13}C_{Atm\,(t+1)} - \delta^{13}C_{Atm\,(t0)})) \tag{1}$$

Where t is the year of observation with t=0 equal to 1840, and R is the turnover time for the FAs in years. Further we assumed no changes in $\delta^{13}C$ of atmospheric $CO_2$ before 1840 and calculated the changes afterwards following the values of atmospheric $CO_2$ of Feng (1998). The soil organic carbon pool size was assumed to be stable over the time and no changes in isotope fractionation during photosynthesis due to a concentration increase in $CO_2$ were taken into account.

We assumed three different turnover times for organic material in the soil of 10,30 and 100 years, which are disccued in soil science (Lichtfouse, 1997; Six and Jastrow, 2002; Wiesenberg et al., 2004).. For a better comparability, both, soils and sediments were Suess-effect corrected, following equation 1, to "before industrialization" values (1840) like it was also done when correcting tree rings for the Suess effect (McCarroll et al., 2009) (for the results please see Fig. S4-7 and Table S4-5). For the highest turnover rate the resulting maximum effect is a depletion of 1.91‰ from 1840 until 2015 for the soils. For the

lake sediments the maximum depletion is 1.71‰ between 1840 and 2010. The older the sediments the smaller the Suess effect. For 100 years turnover time we observe smaller changes for the soils and sediments, of 0.7‰ and 0.63‰, respectively. Despite all uncertainties and simplifications, we can expect that the Suess effect will also affect the FAs deposited in the lake sediments. Thus, it is important to apply this correction also to the sediments, since the terrestrial derived long-chain FAs in the sediment express the $\delta^{13}C$ status of the soil in the year of deposition and have therefore to be corrected in parallel to the soils.

Our results in calculating the Suess effect are considerably higher than previously discussed in the literature from measurements of archived soil samples. A shift of 0.2-0.3‰ in $\delta^{13}C$ (i.e. in the range of the measurement precision) was measured in arable temperate soil samples from 1960s compared to 2000s, i.e. during the period when the Suess effect would be most relevant (Wiesenberg, 2004). Congruently, a 0.1-0.3‰ shift related to the Suess effect was measured for a tropical soil with an estimated turnover rate of <10 years by Bird et al. (2003). The considerably lower measured effects of the Suess

effect compared to our calculated effects might be due to either (i) the ploughing of arable soils or non-permanent grasslands which results in a mix of young and old organic matter, or (ii) considerably lower turnover rates in soils than we assumed for our calculations.

However, with the new findings in our study and others (Petrisic et al., 2017; van Bree et al., 2018) the production of long-chain FAs within the lake has to be considered (see above for discussion). In this case, the Suess effect has a direct link to the

autochthonous produced FAs and a correction for the resulting growing depletion with time should be carried out if the ratio between autochthonous produced and soil derived organic matter would be known. Since the responsible algae, bacteria or microorganisms are not yet identified and the $\delta^{13}C$ source signal is not yet known, a serious correction for the Suess effect is not possible. Therefore a correction for the Suess effect in this study is very speculative (please note that we document the Suess effect corrected values in the supporting information Figure S4-7 and Table S4-5).


#### 4.2.4 Eutrophication, lake and catchment history, in the light of the CSIA

As the data indicate that the C24:0 signal is the most affected of the 3 considered homologues, the following discussion will focus on the C26:0 vs. C28:0 signals (Fig. 4), which were also the homologues allowing the best distinction between the land-uses in the source soils (Sect. 4.1.).

The C26:0 vs. C28:0 CSIA allows a distinction of different units (Fig. 4): before 1900; 1900 to 1940's; 1940's to early 1960's, early 1960's to early 1970's, early 1970's to today. These units confirm the land use changes along different time periods discussed in previous studies led in the lake (e.g. based on diatoms succession, bulk carbon isotopes, eutrophication history;

Lotter, 1998; Teranes and Bernasconi, 2005; Stadelmann et al, 2002, respectively), which attests to the reliability of the CSIA signal to discuss the lake and catchment history.

The oldest sediment samples are deposited prior to the eutrophication start, which beginning was dated from 1885 from (1) phosphorus concentrations inferred from the diatom assemblages and (2) the appearance of varves in the sediment sequence (Lotter et al., 1997; Lotter, 1998). At the onset of the 20th century, a deviation in the C26:0 CSIA data towards lower values is recorded (Fig. 4a) while simultaneously a first important step in eutrophication is reached. Indeed, at that time, the microbial biomass increases (Teranes and Bernasconi, 2005) and a change in diatoms assemblage is recorded (Lotter, 1998), in response to the important industrial development of the catchment and the associated massive wastewater inputs into the lake.

In the early 1940's, a strong shift towards higher values is recorded in the C26:0 CSIA data signal (Fig. 4a). The lake then enters in a severe eutrophication period, marked by an increased influence of the bacterial communities (Neunlist et al., 2002; Teranes and Bernasconi, 2005). Lake water is anoxic below 40 m depth (Niessen and Sturm, 1987). The influence of the land-use changes on the lake response deserves consideration. Indeed, as a result of the Wahlen Plan, a Swiss food self-sufficiency program launched at the beginning of the Second World War, arable lands expand at the country scale (Popp, 2001). In Lake Baldegg catchment, surfaces dedicated to open lands are multiplied by a factor of 3.6 between 1934 and 1945; they even increase by a factor of 4.1 for the cereals (Federal Statistical Office, 1949). Maize is introduced in the catchment during the 1940s, but its dedicated surface is under 3 ha in the mid-1940's and remains small until the 1980s (Federal Statistical Office, 1949; Lotter, 2010). No other cereal is introduced, but the relative proportion of winter wheat strongly increases (Federal Statistical Office, 1949). The agricultural intensification is reflected in the decline of grassland species, the decrease of ruderals of poor soils, the increase of *Urtica* and the appearance of *Ambrosia*, the latter testifying to soil destructuration (pollen analyses of van der Knaap, 2000; Ducerf, 2017). According to air photographs, forest composition also changes to include more coniferous trees, and forest roads develop. Besides, agricultural intensification leads to intense river corrections: for instance, on the Western part of the catchment, 4 small rivers are buried in the 1940s. Such corrections, accompanied by the development of drainage system, will continue until the 1960s.

The isotopic excursion begins in the early 1960's (Fig. 4a), as the lake tends towards its most severe hypertrophic conditions, with a hypolimnion anoxia from 10 m depth (Niessen and Sturm, 1987). The strongly increasing phosphorus concentration fosters the development of photoautotrophic biomass, while the chemautotrophic bacterial biomass is still largely present in the lake, though declining (Teranes and Bernasconi, 2005). This anoxic phase is synchronous to increased sewage sludge inputs, as well as to a strong intensification of pig breeding in the catchment.

This isotopic excursion ends with the introduction of wastewater treatment plants in the catchment (Stadelmann et al., 2002). Later, the artificial oxygenation system set up in the lake in 1982 allows the return to oxic conditions at the bottom of the lake. This favors the development of phytoplanktonic producers, at the expense of the chemautotrophic biomass (Teranes and Bernasconi, 2005).

It is worth noting that from the mid-1940's, all the lake samples (except the early 1960's to early 1970's isotopic excursion) fall into the source soil polygon (Fig. 4a), suggesting that these samples are not, or very little affected by depleted organic material from the unknown source. All the CSIA data of these samples from the forest / arable land / orchard areas fall into the polygon of the source soils signatures. While the sediment contribution from the arable lands can be explained by its associated discontinuous land cover and the agricultural practices (ploughing), the contribution of the forest pool is more surprising. However, most of the forests develop on steep slopes in the catchment, favoring the export of forest soil material towards the lake. Besides, sedimentary inputs into the Lake Baldegg occur mainly during high flow events, which CSIA

imprints were also shown to be dominated by forest contribution in a nearby catchment (Enziwigger catchment; Alewell et al., 2016). Furthermore, the development of trees and shrubs along the streams and on the shores of the lake since the 1940s (air photographs, pollen analysis; van der Knaap, 2000; field observation) may contribute directly to the signal.

### 4.2.5 General considerations

While the units defined with the CSIA match well with the eutrophication and the catchment history, it is remarkable that the oldest sediments (older than 1940) seem to be more affected by depleted material than the younger ones (except the isotopic excursion of the mid-1960's to mid-1970's). Indeed, the maximal extent of the chemautotrophic biomass activity takes place during the most severe eutrophication periods of the lake, i.e. after 1940. But this could also be explained by a change of the presence or in abundance of the in-situ producers of long-chain FAs accompanying with the change of the lakes trophic status.

It is also worth noting that while C24:0 and C26:0 are more depleted than C28:0 for the oldest lake sediments, the opposite is observed for the mid-1960's to mid-1970's excursion. Changes in the microbial biomass composition, resulting in contrasted effects on the *n*-FAs isotopic signature, can be suspected. Here the example of van Bree et al. (2018) can be consulted, as they found a compound-specific enrichment of mostly C28:0 FA in the water column. Also, the accompanying depletion of, in their case, C28:0 FA compared to the terrestrial C28:0 FA signal is giving strong evidence for an aquatic source. However, we

conclude, that the modelling of the different source contributions for the lake sediments cannot be conducted in this study. Any approach to deduce the isotopic signal from the data, without identifying the actual sources and their isotopic signature would be very speculative (see Fig. S3) and would not lead to serious and concise results. Therefore, future research should focus on identification of the aquatic producers of long-chain FAs in lacustrine environments and its isotopic composition.

### 4.3 Triterpenoid biomarkers

The occurrence of cyclic highly specific biomarkers was checked both in soils and lake sediments. Pentacyclic triterpenes such as some triterpenyl acetates, tricyclic diterpenes and methoxyserratenes (biomarkers of Asteraceae, conifers, Pinaceae, respectively; Lavrieux et al., 2011; Otto and Wilde, 2001; Le Milbeau et al., 2013) were investigated. While some non-specific molecules of these families have been identified in soils under the expected land-uses, and some triterpenoids were detected in the lake sediment, the most specific of them were totally absent from the latter. The concentration of these HSB in sediments

is usually lower than the more common linear compounds such as *n*-fatty acids (e.g. Lavrieux, 2011). Accordingly, their non-detection in the Lake Baldegg archive can be due to small undetectable inputs from the catchment or a signal dilution into autochthonous (lake organisms) contribution. Besides, a possible degradation of these pentacyclic triterpenes after their deposition can be hypothesized although the successful use of these molecules for palaeoenvironmental studies suggest their high preservation potential (e.g. Lavrieux, 2011; Guillemot et al., 2017 for triterpenyl acetates; Simoneit, 1986; Stefanova et

al., 2002 for tricyclic diterpenes).

However, in all lake sediment samples, two trimethytetrahydrochrysenes (TTHCs) were detected: 3, 4, 7-trimethyl-1, 2, 3, 4-tetrahydrochrysene (TTHC2) and 3, 3, 7-trimethyl-1,2,3,4-tetrahydrochrysene (TTHC3). These polycyclic aromatic hydrocarbons (PAH) of natural origin derive from the rapid diagenesis of ubiquitous pentacyclic triterpenoids of the oleanane- and ursane series synthesized by upper plants (e.g. Wakeham et al., 1980). These TTHCs were reported during the last decades

in recent lakes sediments (e.g. Wakeham et al., 1980; Yunker and MacDonald, 1995; Jacob et al., 2008), as well as in deltaic environment (Bouloubassi and Saliot, 1993). Their formation in anaerobic conditions via microbial activity (Wakeham et al., 1980) was confirmed by the laboratory experiment of anaerobic transformation of triterpenes into PAH by Lohmann et al. (1990). Despite their production conditions are known, it is still under debate where this transformation takes place and would depend on the study site context: the TTHCs would be synthesized either in leaf litter or in deep soils (Wakeham et al., 1980;

Jacob et al., 2008), during transport (Bouloubassi and Saliot, 1993), or produced *in-situ* in the lake sediment column (e.g. Bouloubassi et al., 2001; Yunker and MacDonald, 2003).

While our investigations revealed the occurrence of these TTHCs in lake core sediments, they were neither detected in the upper soils, nor in river suspended sediments from Lake Baldegg catchment (unpublished results). Hence, the formation of
TTHCs in soils and during transport appears here very unlikely, although their presence in deep soils (as reported by Wakeham et al., 1980) and their subsequent transport through deep soil erosion cannot be fully excluded.

The temporal evolution of TTHCs concentration is provided in Fig. 5. The lowest concentrations are recorded in the earliest part of the archive, before the onset of the eutrophication, and increase as the latter start. The maximal concentration is reached in the middle of the 1960's, i.e. synchronously to the isotopic excursion recorded in CSIA. The evolution of TTHCs
concentration was confronted to ratios of $\delta^{13}C$ FAs ($\delta^{13}C$ C24:0/$\delta^{13}C$ C26:0; $\delta^{13}C$ C26:0/$\delta^{13}C$ C28:0; $\delta^{13}C$ C24:0/$\delta^{13}C$ C28:0). As expanded above (Sect. 4.2.), a high discrepancy in isotopic values between long-chain FAs of close chain-length points to a degradation of the isotopic signal. Then, the more the values differ (i.e. the more the ratio of their isotopic values is >1 or <1), the more the isotopic signal of one of the FAs can be considered as degraded. Keeping in mind that such a ratio is not an absolute indicator because of some variability results from the biosynthesis pathway, one can still consider the overall
evolution of the ratio along the core. C28:0 being considered as only little affected by authochtonous production (Sect. 4.2.), $\delta^{13}C$ C26:0/$\delta^{13}C$ C28:0 and $\delta^{13}C$ C24:0/$\delta^{13}C$ C28:0 ratios are taken as more reliable than the $\delta^{13}C$ C24:0/$\delta^{13}C$ C26:0 ratio.

Interestingly, the TTHCs concentration evolution is highly similar to the $\delta^{13}$FAs ratios trend, even more for the $\delta^{13}C$ C24:0/$\delta^{13}C$ C28:0 than for the $\delta^{13}C$ C26:0/$\delta^{13}C$ C28:0. This suggests a TTHCs concentration under the control of the lake bacterial activity, similarly as the CSIA signal. In other words, the TTHCs signal archived in the Lake Baldegg sediments most probably testifies
to an in-situ degradation of pentacyclic triterpenes, consequent to the bacterial activity favored by the anoxic conditions in the water column (Wakeham and Canuel, 2016). While these compounds have successfully been used in many contexts for palaeoenvironmental reconstructions (e.g. Lavrieux, 2011; Dubois and Jacob, 2016; Guillemot et al., 2017), our results show the impossibility to use them to decipher the terrestrial inputs in the case of the highly eutrophic and microorganism-dominated Lake Baldegg environment.

Thus, the microbial activity masks to a large extent the terrestrial molecular inputs in the Lake Baldegg, and affects the linear compounds (as shown by the CSIA) as well as the cyclic ones (as shown by the HSB).

## 5 Conclusions

The aim of this study was to apply a mixed CSIA and HSB approach to the highly eutrophic context of Lake Baldegg catchment. The main land-uses were successfully discriminated with the CSIA but align along a line. The CSIA signals of
arable lands as well as orchards plot halfway between grasslands and forests, which may render difficult to correctly attribute the sources of sediment samples lying between grasslands and forests end-members. Most of the recent lake sediments plot within the forest soil pool, underlining the potential important contribution either of the steeply sloping and loosely structured forest soils or to tree lines growing along the streams and around the lake, which could contribute directly to the signal transported to the lake sediment archive. Further studies are required to investigate the extent of this potential contribution.
However, all lake sediments older than 1940's, as well as those from mid-1960's to mid-1970's actually fall out of the polygon of today's source soils signatures.

Although the influence of not yet identified sources to our lake sediments, as expanded above, is very likely, the fact that the C28:0 lake isotopic values fall in the same range as the soils tends to indicate (1) that the soils are most probably the main sources of C28:0 FA and (2) would hint into the direction of an additional source with low C28:0 FA concentrations compared to C24:0 and C26:0 FAs. This leaves a number of potential additional sources, namely, (i) a larger contribution of

Sphagnum/moss derived organic matter released from historical peat bogs or the riparian zone as described above, or during time periods of hypereutrophic status of the lake (ii) in-situ FA production of preferably C24/26:0 FA by either methanotroph bacteria or other bacteria which were using $\delta^{13}C$ depleted $CO_2$ derived from methane-oxizing bacteria (MOB), (iii) algae with depleted $\delta^{13}C$ values due to described effects of hydoxilation reaction of $CO_2$ combined with high pH values in the epilimnion and $CO_2$ undersaturation, or (iv) due to the uptake of MOB derived $CO_2$ depleted $\delta^{13}C$ analogous to the bacteria mentioned

before.

While the long-chain fatty acids are becoming widely used for CSIA as markers of the terrestrial sediment contribution to rivers and lakes, our results underline the need to temper this standpoint. Some lacustrine macrophytes, bacteria and microbial organisms were previously shown to produce also long-chain FAs, and also our study highlights that an interference of the terrestrial isotopic signal linked to aquatic activity might be underestimated. CSIA was proven to be not suitable to

quantitatively unmix terrestrial sources from the Lake Baldegg historic sediments, and thus to apportion the relative contribution of different land-uses to the sedimentary archive as long as the isotopic signal of the missing source is not known.

While the isotopic signal, especially C24:0 and C26:0 FAs until 1940 and C28:0 FA during the 1960s and 1970s, is clearly influenced by unknown, but most likely aquatic sources, land-use and catchment historical development are still surprisingly accurately reflected in the background patterns: human activities and land-uses directly impacted the trophic level of the lake

and its accompanying biomass, imprinting its mark on the FAs isotopic signal. The main phases of land-uses and catchment history over the last 150 years are thus still visible in the CSIA results. More than affecting just linear compounds, it is highly probable that microbial activity also affected the more specific cyclic molecule assemblages, as testified by the presence of in-situ produced TTHCs. Special care should thus be taken for further studies on eutrophic systems, where a strong bacterial activity is known or suspected.

To conclude, we see the imprints of plants AND bacteria in the Lake Baldegg sediments: the influence of terrestrial derived compounds and their changes with land use management are very precisely documented, but at the same time the influence of in-situ produced FAs is very likely and has an impact on the overall isotopic composition of long chain FAs..

**Acknowledgements.** This study was funded by the European COST Action ES1306, "Connecting European connectivity research" and was finalized in the framework of the IAEA Coordinated Research Project (CRP) "Nuclear techniques for a

better understanding of the impact of climate change on soil erosion in upland agro-ecosystems" (D1.50.17). Core recovery was financially supported by an ETH research grant (CH1-02-08-2). We wish to thank Dr. Stefano Bernasconi (ETH Zurich) for granting access to the ASE, and Judith Kobler-Waldis, Thomas Kuhn, Simon Imhof, Oliver Rehmann and Lukas Burgdorfer for their help in the laboratory. We thank Robert Lovas, Pius Stadelmann and Franz Stadelmann (Canton Lucerne) for providing helpful information about the catchment. Further we thank Stefano Crema and Marco Cavalli for their support

to calculate the connectivity index. Our acknowledgements are also addressed to the land owners for sampling permissions and their curiosity about our work.

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

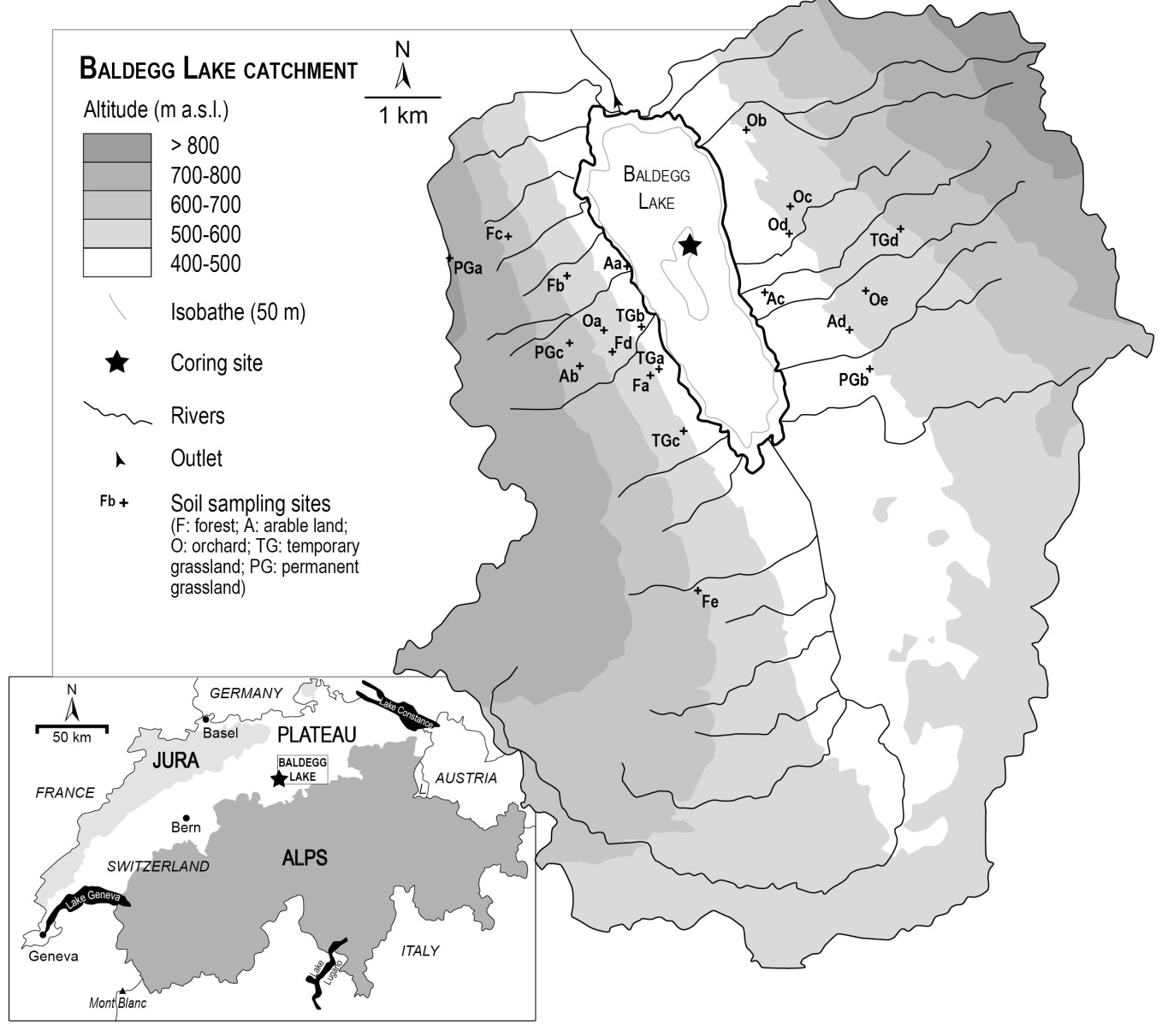

**Figure 1: Catchment of Lake Baldegg with sampling sites.**

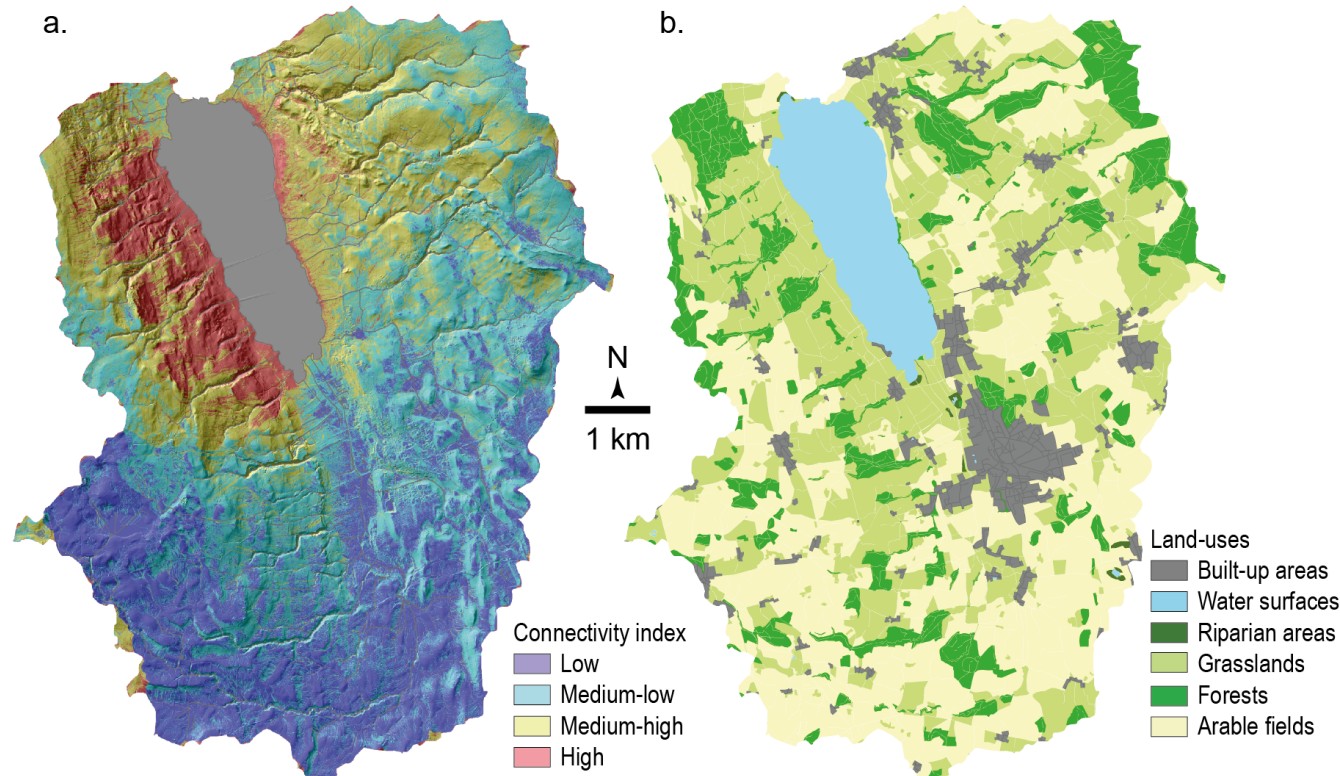

**Figure 2: a. Connectivity index for the Lake Baldegg catchment, with a topographic map underlying. The lake is indicated in grey. b. Land-use map.**

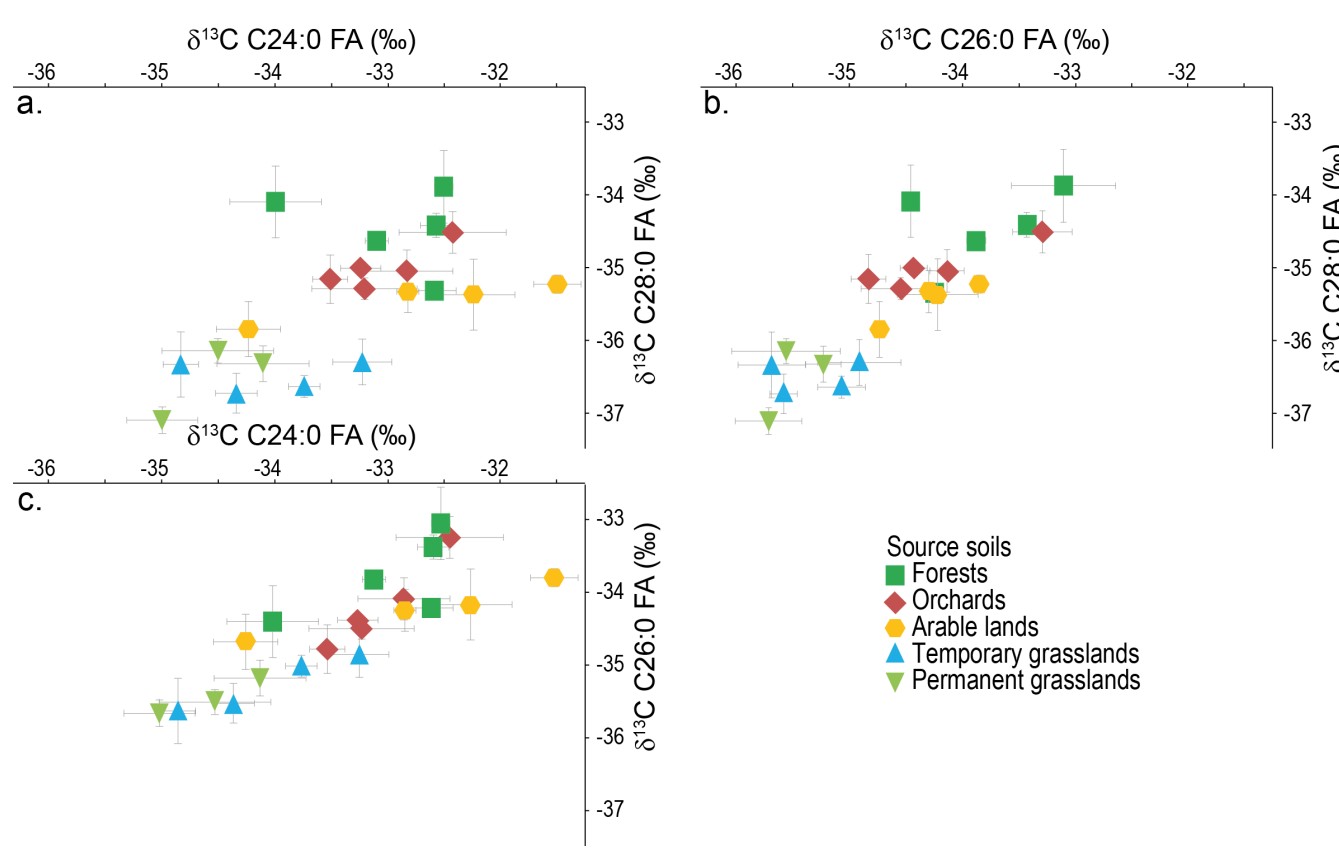

**Figure 3: δ¹³C of the FAs (a.) C24:0 vs. C28:0, (b.) C26:0 vs. C28:0, (c.) C24:0 vs. C26:0 in soils. Error bars: standard deviation of the triplicate measurements.**

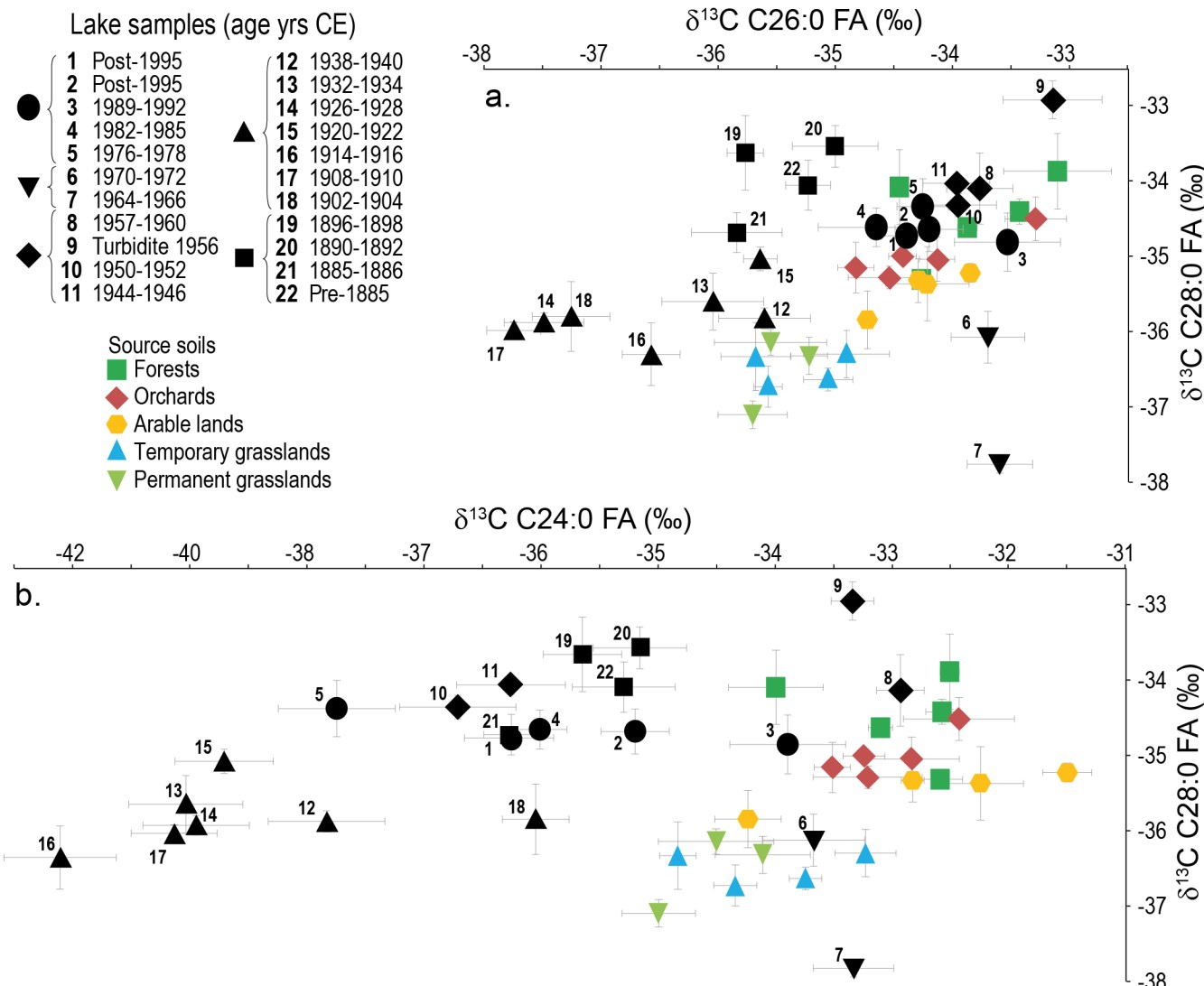

**Figure 4:** δ¹³C of the FAs (a.) C26:0 vs. C28:0, (b.) C24:0 vs. C28:0, in lake sediments, compared to soils. Note the different scale for the x axis between (a.) and (b.). Error bars: standard deviation of the triplicate measurements.

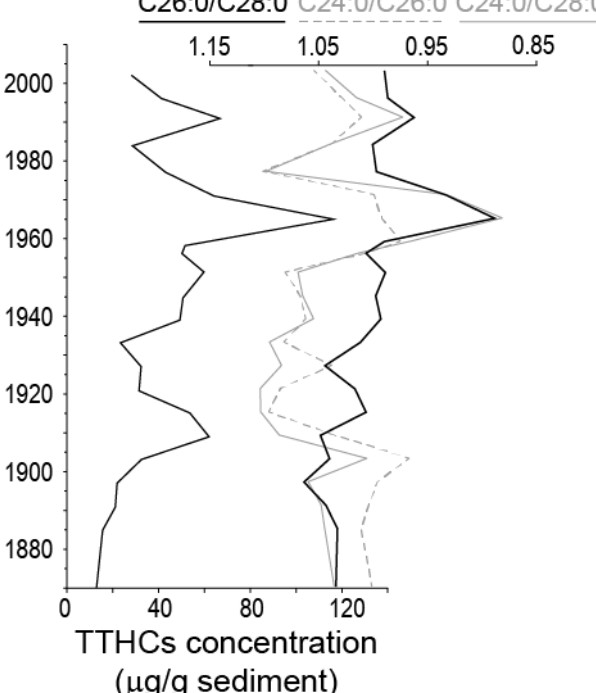

**Figure 5: Evolution of the TTHCs concentration (sum of TTHC2 and TTHC3) along the sediment core, compared to evolution of the ratios of δ¹³C of FAs,**