# Peer review of "Plants or bacteria? 130 years of mixed imprints in Lake Baldegg sediments (Switzerland), as revealed by compound-specific isotope analysis (CSIA) and biomarker analysis"

_Biogeosciences, 2018_

## Referee Comment (RC1) · Anonymous Referee #1 · 27 Jul 2018

The authors state in P3 L 31-32. For the forest soils, the humus layer was removed prior to sampling.

I find this counter intuitive - as this layer contain most (recent) organic matter and in term of connectivity is most likely to contribute material (thus lipid markers) to the lake. It be useful for the authors to provide an reasoning why they think this removal was valid in the context of the paper and its aims

---

## Short Comment (SC1) · 31 Jul 2018

Thank you for this question! In the case of the humus layer of the Baldegger Lake forest soils we only find Oi and Oe horizons (e.g. partly degraded and intact plant material) but no fully fermented Oa horizon (humus material which might be eroded with the soil sediments). Lake sediments were carefully prepared and all recognizable plant residues were picked out prior to lipid extraction. Thus, we think that with this method we are able to minimize the influence of the forest humus layer (e.g., in this case non fermented plant input) on the lake sediments. Sorry for the confusion, we

will describe this clearly in a revised manuscript. However, in a currently ongoing study we analysed the isotopic composition oft he FAs in the different humus horizons at a forest of Baldegger lake and can show that they are no option of being a major reason for the deviation of the lake sediments from the mixing polygon. Fresh leaves, litter and partially fermented material are actually heavier than the top soil (Ah) (0.1 - 1.1 permil for a spruce site and up to 1.7 permil for a maple tree site) and can thus not explain the deviation from the mixing polygon in the isotopically lighter direction. Thus, regarding the deviation of the lake sediments from the mixing polygon during the older time intervals covered by the lake sediment core, we don't think that the humus layer of the forest could be the crucial factor as a "missed source". Especially since we could then assume that the humus layer would also have had an affect on the recent lake sediments, but they plot within the mixing polygon. Kind regards, Axel Birkholz Christine Alewell

---

## Referee Comment (RC2) · Anonymous Referee #2 · 7 Aug 2018

Comments and over view: The manuscript is well written and describes a study of the sedimentation changes in a hypertrophic freshwater lake over a period of 130 y before present. Most of the techniques used in this study were appropriate and it is good to see a study trying to use an alternative biomarker to the CSIA of fatty acids as a cross check to validate those results. However, there are some issues in this manuscript that I feel need addressing. My comments are therefore on the overall concept presented in the manuscript and are intended to be helpful. I focus on 1) the need to apply known corrections to data and 2) the three hypotheses raised by the authors to interpret their

results.

1) It has been documented (Verburg 2007) that samples from lake sediment cores going back in time to when the isotopic value of $\delta$13C in the atmospheric CO2 was less depleted than at present, need to be corrected for this Suess effect. This is because plants assimilate atmospheric CO2 during photosynthesis to produce the fatty acids. Those isotopic values on that day are transferred to the fatty acids which bind to the soil and which are measured in the CSIA tracer study. Consequently, to use present day land use samples as the reference sources for the sediment in the core samples over time, the core data must be corrected for the Suess effect. The authors argue that, compared with isotopic depletion of the tracers in the deeper sections of the core, the Suess effect is "considered as negligible" and was not done. My understanding is that the main objective of science is find the truth from the limited resources and knowledge available. The Suess effect is a known truth and needs to be applied to the core data. It doesn't matter that effect is small relative to other unknowns, I believe that the authors should apply the correction to the data. This will save the authors a page of text arguing that it is not needed.

My contention is that, if everyone picks and chooses which corrections to apply to specific types of data, the supplementary data provided with each manuscript will be worthless.

2) The most important finding in this study is that there is increasing isotopic depletion in the FAs with depth and the authors have correctly associated this effect with methanogenesis by bacteria in the anaerobic sediments. This isotopic depletion, however, causes a problem where the sediment sample isotopic values do not plot within the source polygons, as in this study. The authors raise three hypotheses suggesting that "either (1) that values have to be corrected for the atmospheric 13C-depletion of the industrial era (Suess effect; Suess, 1955; Keeling, 1979) (2) that the major contributors to most of the lake FA isotopic signal are not the main source soils of the catchment, or (3) that the signal, originating from catchment soils, was altered after its

introduction into the lake."

Simply applying the Suess correction removes hypothesis 1.

Hypothesis 3) violates the primary assumption of the CSSI sediment tracing technique (Gibbs 2008) that the isotopic signatures of FAs bound to the soil particles do not change over time. The discussion around this point is speculative and then assumed to be correct without supporting evidence. The authors need to provide irrefutable evidence for the alteration in the isotopic signature of a soil-bound FA after it has been deposited in the sediment or acknowledge the basis for using FAs as tracers as correct, which removes this hypothesis.

This leaves hypothesis 2) "that the major contributors to most of the lake FA isotopic signal are not the main source soils of the catchment". This is a realistic hypothesis, which the authors need to investigate further. Internal cycling is common in most lakes. Expanding on this concept, the 'other' sources need to be able to consume a food that is isotopically depleted and the consumer must be able to produce long-chain, even-carbon number FAs that can bind to the sediment particles rendering them stable against decomposition and fractionation to short-chain molecules. Methanotrophic bacteria assimilate the methane and produce odd-carbon chain lengths mostly in the C15, C17 and C19 chain lengths. Consequently, the bacteria are not the source of the even-carbon FAs, but they are the food source for chironomid larvae living in the anoxic sediment. Work by Jones and Grey (2004: Boreal Environment Research 9: 17–23), Deines et al (2007: Aquatic Microbial Ecology 46: 273–282), and others, indicate that chironomid larvae can take up the depleted isotopic signature from the bacteria and acquire highly depleted isotopic signatures. They may leave their skins and head capsules in the sediment when they hatch or the unseived sediment samples may have contained whole organisms. In severely hypertrophic lakes, chironomid populations can reach very large numbers. And yes, chironomids produce FAs (e.g., Makhutova et al 2017: Contemporary Problems of Ecology, 10: 230–239).

A further issue around the selection of the fatty acid tracers, the authors have chosen to use only C24:0, C26:0 and C28:0 and wonder why they cannot discriminate grasses. Grasses produce very low levels of these long-chain FAs but high levels of C18:0 and C18:1 fatty acids. There is also no bulk $\delta$13C data. Including these tracers may help sort out some of the source identification problems in this manuscript. If all the tracers are not included in the isotopic inputs to the mixing model, they cannot be seen in the model output.

In the conclusions, the authors have reiterated a statement "strongly overprinted by carbon exchanges". As stated above, this is an unsupported hypothesis which cannot be correct. It is more likely that the FA signatures from the terrestrial sources have been overprinted by an in-lake source that has not been sampled and more work is required to identify that source.

---

## Short Comment (SC2) · 28 Aug 2018

Thank you for your valuable comments and suggestions. Regarding point 1, we agree with the reviewer to add a correction of the lake sediment isotopical data for the Suess effect. In a revised version we propose to show the corrected values next to the original ones. The correction also has to be applied to the source soil samples. However, the sampled soil organic matter consists of a mixture of different (unknown) ages, from recent production to a few hundred years. A side by side comparison of values with and without Suess correction might eliminate our hypotheses to explain the depleted

d13C values of the fatty acids in the older sediments with the Suess effect.

Regarding point 2.1: We thank the reviewer for his possible explanation of our observation. Marlène Lavrieux, who did all the preparative work with the lake sediment samples has experience in the work with chironomid larvae and is convinced that there were neither complete organisms nor residues of chironomid larvae present in the lake sediments. Further we checked the cited literature (Makhutova et al. 2017) and could only find documentation of production of saturated long-chain fatty acids up to chain length of 22 C-atoms by chironomidae. Is there any reference we might have overlooked where it is shown that chironomid larvae are able to produce fatty acids longer than C22:0 in significant amounts?

Due to the extreme eutrophication history of the lake and changing bacterial communities we cannot exclude that there might have been bacteria present with the ability to produce long-chain fatty acids. Depletion of long-chain fatty acid 13C values in marine sediments was described as being a possible result of a mixture of terrestrial produced fatty acids and depleted in-situ produced bacterial long chain fatty acids in an anoxic environment (Gong and Hollander, 1997; Teece et al., 1999 (d13C depletion of short/mid-chain fatty acids after production under anoxic conditions by marine bacteria), Fang et al., 2014). In a revised version of the manuscript we will discuss this hypothesis in more detail. But this would not really change the conclusion of our study: that the activity of bacteria would alter our isotope signals.

However, we discussed further options and we will also include a discussion about common reed (phragmites australis) in the riparian zone as a possible additional source. Photosynthesis of Phragmites with CO2 derived from oxidized methane could be an optional source for depleted long-chain fatty acids (cf. Alewell et al. 2011, d13C depletion of mosses, induced by photosynthesis with methane derived CO2, effected the bulk carbon d13C in Scottish bog). It is well possible that along with the intensification of agriculture in the catchment the presence of phragmites in the riparian area was decreasing. This could be an explanation for the depleted values until world war two.

With the absence (very low stock left) of phragmites also the influence of a depleted source of the fatty acids would have disappeared.

Regarding point 2.2: We choose the long-chain fatty acids C24:0-C30:0 because they are mainly produced by higher plants and other known producers (bacteria cf. literature in the manuscript) were not identified in Baldegger lake so far. Short- and mid-chain fatty acids are also produced by bacterial and aquatic organisms. In a lake environment we expect increased production of these sources for the short- and mid-chain fatty acids in addition to the higher plants, which would heavily impede the sediment apportionment. Therefore, we concentrated on the long chain ones. 2.3: In the same study also suspended sediments were sampled and the sample amount was not enough to do additional analysis for bulk carbon. The same was true for the lake sediment samples. Due to the high temporal resolution, all material was used for lipid extraction. Kind regards, Axel Birkholz, Marlène Lavrieux and Christine Alewell.

C.R. Gong, D.J. Hollander, Differential contribution of bacteria to sedimentary organic matter in oxic and anoxic environments, Santa Monica Basin, California Org Geochem, 26 (1997), pp. 545-563.

M.A. Teece, M.L. Fogel, M.E. Dollhopf, K.H. Nealson Isotopic fractionation associated with biosynthesis of fatty acids by a marine bacterium under oxic and anoxic conditions Org Geochem, 30 (1999), pp. 1571-1579.

Jidun Fang, Fengchang Wu, Yongqiang Xiong, Fasheng Li, Xiaoming Du, Da An, Lifang Wang, Source characterization of sedimentary organic matter using molecular and stable carbon isotopic composition of n-alkanes and fatty acids in sediment core from Lake Dianchi, China, Science of The Total Environment, Volumes 473–474, 2014,Pages 410-421.

Alewell, C., Giesler, R., Klaminder, J., Leifeld, J., and Rollog, M.: Stable carbon isotopes as indicators for environmental change in palsa peats, Biogeosciences, 8, 1769-1778.

---

## Author Response (AR1)

**Point by point reply to reviewer comments**

**Reviewer comments in bold text.**

**Anonymous Referee #1**

**The authors state in P3 L 31-32. For the forest soils, the humus layer was removed prior to sampling.**
**I find this counter intuitive - as this layer contain most (recent) organic matter and in term of connectivity is most likely to contribute material (thus lipid markers) to the lake. It be useful for the authors to provide an reasoning why they think this removal was valid in the context of the paper and its aims.**

In the case of the humus layer of the Lake Baldegg forest soils we find for four out of five locations humus profiles with partly degraded (Oe) and intact plant material (Oi), and only one location with a fully fermented Oa horizon (fine humus material which might be eroded with the soil sediments). Lake sediments were carefully prepared and all recognizable plant residues were picked out prior to lipid extraction. Thus, we think, that with this method we are able to minimize the influence of the forest humus layer (e.g., in this case non-fermented plant input) on the lake sediments.

We added a more detailed description of the humus constitution of the forest soils on
**P 3 L33-36** in the marked manuscript
*For all the sites the humus layer was removed prior to sampling.*
*At four of five investigated forest sites no Oa layer was present (only partly degraded material (Oe) and intact plant material (Oi)). Only at one site (Norwegian spruce and Thuidium tamariscinum moss) an Oa layer had built up. For all the sites the humus layer was removed prior to sampling.*

In an ongoing study we analysed the isotopic composition of the FAs in the different humus horizons of the Lake Baldegg forest soil with the Oa horizon. The values of the Oa horizon FAs were slightly depleted in $\delta^{13}C$ compared to the Ah horizon, but also the depleted values lie within the area of the forest isoplot, and wouldn't be separable as a discrete source.

We added a section in the reworked results section on
**P 7 L13-23** in the marked manuscript
*The input of organics from humus material and mixing into the lake sediments might be discussed as a potential additional source. However, great care was paid to remove any macroscopic organic material from the sieved soil and lake sediment samples. Even if we missed some highly decomposed Oa material, a study about fractionation processes of FAs in the humus layer of forest soils at Baldegg Lake catchment recorded no or only slight changes in the isotopic signal from Oa to Ah horizon (unpublished data). For our site with the Oa horizon, C28:0 and C30:0 FA were only slightly depleted by 0.2 - 0.3‰ compared to the Ah horizon. C24:0 and C26:0 were depleted by 0.8 and 1‰ respectively. But these humus $\delta^{13}C$ values are -33.8‰ for C26:0 FA and -34.6‰ for C28:0 FA and can thus not explain shifts of C26:0/C28:0 to values more negative than -36‰ (compare Fig. 4). Further, these humus $\delta^{13}C$ values still lie in the isotopic range of the five analyzed forest locations (Fig. 4) and would not be separable as a discrete source. Also, as today's isotopic signals of lake sediment samples plot within the polygon of the source soil signals, we rather expect a source or process different from today's conditions being the cause for the deviation of isotopic signals.*

**Anonymous Referee #2**

**Comments and over view: The manuscript is well written and describes a study of the sedimentation changes in a hypertrophic freshwater lake over a period of 130 y before**

**present. Most of the techniques used in this study were appropriate and it is good to
see a study trying to use an alternative biomarker to the CSIA of fatty acids as a cross
check to validate those results. However, there are some issues in this manuscript that
I feel need addressing. My comments are therefore on the overall concept presented
in the manuscript and are intended to be helpful. I focus on 1) the need to apply known
corrections to data and 2) the three hypotheses raised by the authors to interpret their results.**

**1) It has been documented (Verburg 2007) that samples from lake sediment cores
going back in time to when the isotopic value of _13C in the atmospheric CO2 was less
depleted than at present, need to be corrected for this Suess effect. This is because
plants assimilate atmospheric CO2 during photosynthesis to produce the fatty acids.
Those isotopic values on that day are transferred to the fatty acids which bind to the
soil and which are measured in the CSIA tracer study. Consequently, to use present
day land use samples as the reference sources for the sediment in the core samples
over time, the core data must be corrected for the Suess effect. The authors argue
that, compared with isotopic depletion of the tracers in the deeper sections of the core,
the Suess effect is "considered as negligible" and was not done. My understanding
is that the main objective of science is find the truth from the limited resources and
knowledge available. The Suess effect is a known truth and needs to be applied to the
core data. It doesn't matter that effect is small relative to other unknowns, I believe that
the authors should apply the correction to the data. This will save the authors a page
of text arguing that it is not needed.**
**My contention is that, if everyone picks and chooses which corrections to apply to
specific types of data, the supplementary data provided with each manuscript will be
worthless.**

We did a correction for the Suess Effect of lake sediments and catchment soils and calculated their
corresponding values for 1840. With that method it was possible to compare all the sediments and
soils at the same time. Assuming a very high incorporation rate of organic matter into the soil and the
even more distinct depletion curve of straw by Zhao et al. (2001), the maximum depletion with time
was 0.22‰ for the soils and 0.16‰ for the lake sediments. These values are smaller than the
analytical uncertainty and since these values are based on the estimation of paramaters (e.g. soil
incorporation rate), which in our case were actually chosen to rather overestimate the effect (high
incorporation rate, depletion curve of straw) we choose only to show the corrected values and compare
them with the original ones.

We added a new section "4.2.1 The Suess effect as a possible influence on CSSI of lake sediments" on
**P6 L14 – P7 L 5** in the marked manuscript
And we added table **S4** in the supplementary material.

**2) The most important finding in this study is that there is increasing isotopic depletion
in the FAs with depth and the authors have correctly associated this effect with
methanogenesis by bacteria in the anaerobic sediments. This isotopic depletion, however,
causes a problem where the sediment sample isotopic values do not plot within
the source polygons, as in this study. The authors raise three hypotheses suggesting
that "either (1) that values have to be corrected for the atmospheric 13C-depletion of
the industrial era (Suess effect; Suess, 1955; Keeling, 1979) (2) that the major contributors
to most of the lake FA isotopic signal are not the main source soils of the
catchment, or (3) that the signal, originating from catchment soils, was altered after its
introduction into the lake."**
**Simply applying the Suess correction removes hypothesis 1.**
**Hypothesis 3) violates the primary assumption of the CSSI sediment tracing technique
(Gibbs 2008) that the isotopic signatures of FAs bound to the soil particles do not**

**change over time. The discussion around this point is speculative and then assumed to be correct without supporting evidence. The authors need to provide irrefutable evidence for the alteration in the isotopic signature of a soil-bound FA after it has been deposited in the sediment or acknowledge the basis for using FAs as tracers as correct, which removes this hypothesis.**

We removed this argumentation from the manuscript and only left one last speculation about a potential alteration, but were admitting, that there were so far no studies showing such a process.
**P10 L36** in the marked manuscript
*A diagenetic transformation of the FAs isotopic signal can also be speculated for the sediments older than 1940. Such an assumption would mean that these sediments would have been affected by carbon exchanges years to decades after their deposition, during the most severe eutrophic phases of the lake history. Why these exchanges would not have affected the younger sediments remains unexplained. And so far, no cases of such a diagenetic transformation have been described.*

**This leaves hypothesis 2) "that the major contributors to most of the lake FA isotopic signal are not the main source soils of the catchment". This is a realistic hypothesis, which the authors need to investigate further. Internal cycling is common in most lakes. Expanding on this concept, the 'other' sources need to be able to consume a food that is isotopically depleted and the consumer must be able to produce long-chain, even-carbon number FAs that can bind to the sediment particles rendering them stable against decomposition and fractionation to short-chain molecules. Methanotrophic bacteria assimilate the methane and produce odd-carbon chain lengths mostly in the C15, C17 and C19 chain lengths. Consequently, the bacteria are not the source of the even-carbon FAs, but they are the food source for chironomid larvae living in the anoxic sediment. Work by Jones and Grey (2004: Boreal Environment Research 9: 17–23), Deines et al (2007: Aquatic Microbial Ecology 46: 273–282), and others, indicate that chironomid larvae can take up the depleted isotopic signature from the bacteria and acquire highly depleted isotopic signatures. They may leave their skins and head capsules in the sediment when they hatch or the unseived sediment samples may have contained whole organisms. In severely hypertrophic lakes, chironomid populations can reach very large numbers. And yes, chironomids produce FAs (e.g., Makhutova et al 2017: Contemporary Problems of Ecology, 10: 230–239).**

My colleague, experienced with chironomid larvae was doing the sample preparation and is convinced that there were no complete organisms or residues of chironomid larvae present in the sediments in relevant amounts. But we took up this interesting point and added a section in the completely reworked results part.
**P8 L5-9** in the marked manuscript
*One potential missed source might be the influence of complete organisms or residues of e.g. chironomid larvae on the $\delta^{13}C$ signal of the lake sediment. Fatty acids produced by these larvae might be depleted in $\delta^{13}C$ (Makhutova et al., 2017) and could act as a potential depleted source. However, sample preparation was done in paying attention to the possible occurrence of chironomid larvae, and we can thus exclude the presence of these organisms in relevant amounts. Moreover, no literature was found stating the production of FAs longer than C22:0 by these larvae.*

**A further issue around the selection of the fatty acid tracers, the authors have chosen to use only C24:0, C26:0 and C28:0 and wonder why they cannot discriminate grasses. Grasses produce very low levels of these long-chain FAs but high levels of C18:0 and C18:1 fatty acids.**

We added an explanation in the method section why we choose the above-mentioned FAs.

**P4 L33 - P5 L2** in the marked manuscript
*Only long-chain FAs ≥C24:0 were investigated. These are characteristic for the higher plant input into the soil (Eglinton and Eglinton, 2008). Short- or mid-chain FAs can also be produced by bacteria or aquatic plants and would bias our approach to trace back the terrestrial input into the lake.*

**There is also no bulk _13C data. Including these tracers may help
sort out some of the source identification problems in this manuscript. If all the tracers
are not included in the isotopic inputs to the mixing model, they cannot be seen in the
model output.**

In the same study also suspended sediments (Filter) were sampled and the sample amount was not enough to do additional analysis for bulk carbon. Due to the high temporal resolution, also all material of the lake sediments was used for lipid extraction.
Moreover, by comparing concentration of bulk C and FAs through the humus profile to the upper soil horizon, it was obvious that the FAs are the more recalcitrant/stable biomarker. Concentration per g TOC in the upper soil horizon is up to factor 10 enriched compared to the humus layer. This makes them a much more suitable tracer for terrestrial input than the more labile bulk C.

**In the conclusions, the authors have reiterated a statement "strongly overprinted by
carbon exchanges". As stated above, this is an unsupported hypothesis which cannot
be correct. It is more likely that the FA signatures from the terrestrial sources have
been overprinted by an in-lake source that has not been sampled and more work is
required to identify that source.**

As pointed out above, we removed the unsupported hypothesis.
Regarding the 2$^{nd}$ part of the comment and the new comment of the reviewer, which we received with the Editors decision letter (see below in green), we completely revised the "Results and discussion" part. We are providing two new hypotheses, in-situ production of FAs by algae, or lateral transport from wetlands/peat with depleted $\delta^{13}$C signature. The possible production by methanotrophic bacteria is also part of the "Results and Discussion" section. We did not, as suggested by the reviewer, use the most negative sediment sample as source signal of bacterial contribution. If it is produced by methanotroph bacteria, then the FAs would probably be even more depleted. And since we assume, as stated in the new manuscript, that C28:0 FA is mainly produced by terrestrial higher plants, and the concentrations of the FAs C24:0 and C26:0 (2 new concentration graphs Fig S1 and S2 are added in the supplementary material) in the sediments do not differ much from concentration pattern in the soils, the missing source would have to be more depleted than the lake sediment from 1908-1910 to cause this negative isotopic shift.
We added the two new sections *"4.2.2 The likelihood of missing an additional terrestrial source to the isotopical FA signal of lake sediments"* on **P7 L6** and *"4.2.3 The likelihood of missing an in-situ source to the isotopical FA signal of lake sediments"* on **P8 L4**. And we revised the "Conclusions" and give an outlook how it would be possible to identify the missing source, as you can see in the marked manuscript. Mainly **P12 L21-36** and **P13 L13-L20**. And we added **Fig. S1** and **Fig. S2** in the supplementary material

**2$^{nd}$ comment of Reviewer2 with the Editors decision letter**
Their comment that the discussion will not change their conclusion is also rather dogmatic. The missing information such as bulk delta13C data could have helped their cause. The use of the polygon is to define the sources contributing to the sediment in the mixture. If the mixture values lie outside the polygon, there is a source missing. In this case rather than look for the missing source, the authors have put forward a hypothesis which has the potential to destroy the integrity of the compound specific stable isotope source tracing technique. The most obvious missing source not modelled in their paper is the methanogenic bacteria in the sediment. If the CSIA data from the 1908-1910 sediment depth is used as the bacteria signature, that closes the polygon and all the rest of the sediment samples fall within the new polygon. This is a realistic solution to the

author's problem. They are already postulating an effect caused by these bacteria and I agree that that is the most likely solution. I do not believe that any bacteria has sufficient energy to break the ionic bond between the terrigenous FA and the clay soil particle - in many other studies this bond has remained intact over thousands of years.

Conversely, for fatty acids to bind to the soil, they have to be produced insitu and apparently these methanogenic bacteria can do that. If they are producing the long chain FAs and the terrigenous signatures are not being replaced (as they won't be at that depth in the sediment), the new FA signatures will overwrite them and they will be below detection limit. This is not that "activity of bacteria would alter our isotope signals", which implies breaking the ionic bonds of the terrestrial FA, but simply overwhelming the terrestrial FA signatures with the new bacterial signatures. Expressed in this way, the research is providing new information about the effects of hypereutrophic lake sediments.

[revised manuscript text omitted]
 could have lead to a broader riparian area with wetlands, which have drained themselves into the lake over time or where drained by the farmers to use the fertile riparian area. A lot of organic material would have been transported during such a process. Furthermore, there were reedlands, with *phragmites australis*, until 1944 next to the main inflow at the southern end of the catchment. From 1944 they started cutting of peat and in 1955 they completely drained this reedland. Today it is a small lake, Siedeweiher, with wetlands, still containing *phragmites australis*, surrounding it. Taking only these two informationthese two information into account, another possible explanation for the negative values of the FAs C24:0 and C26:0 could be a larger contribution of wetlands organic matter derived from e.g. *phragmites australis* or sphagnum species, to the eroded sediments. Especially sphagnum species comprise C24:0 and C26:0 FAs as most concentrated fatty acids (Baas et al., 2000, Pancost et al., 2002). Photosynthesis of Phragmites or mosses in the riparian zone or in peats respectively with CO₂ derived from oxidized methane could be an optional source for depleted long-chain fatty acids (cf. Alewell et al. 2011, d¹³C depletion of mosses, induced by photosynthesis with methane derived CO₂, effected the bulk carbon d¹³C in Scottish bog). In the plant-humus study mentioned above (Hirave et al., 2018), we found δ¹³C values for *Thuidium tamariscinum*sphagnum moss FAs, -38‰ - -39.4‰, which were depleted up to 6 ‰ permil compared to Norwegian spruce organic material. However, δ¹³C values of long-chain FAs from a Scottish peat core show are not depleted and range between -29.5‰ and -32.8‰ (Ficken et al., 1998) and would not be an adequate solution for our missing source. However to proveFor visualizing a possible wetland signalwe choose to use the *Thuidium tamariscinum* mosssphagnum δ¹³C values from the Baldegger lakeLake Baldegg catchment to simulate the missing source (see Figure 4neu5). Apparently this signal could explain our most of our lake sediment samples. Even more if we take into account the relative low C28:0 FA concentration in Sphagnum (Baas et al., 2000, Pancost et al., 2002) compared to C24:0 and C26:0 FAs. Also for *Thuidium tamariscinum* moss we observed C24:0 dominating over C26:0 and C28:0 having the least concentration of the three (Hirave et al., 2018 in prep). The larger proportions of C24:0 and Sphagnum C26:0 FA would clearly dominate over the the Sphagnum 
[revised manuscript text omitted]

**Seite 6: [1] Gelöscht**            **Christine Alewell**            **23.11.18 10:50:00**

**Seite 6: [1] Gelöscht**            **Christine Alewell**            **23.11.18 10:50:00**

**Seite 6: [2] Gelöscht**            **Microsoft Office-Benutzer**            **27.11.18 07:43:00**

**Seite 6: [2] Gelöscht**            **Microsoft Office-Benutzer**            **27.11.18 07:43:00**

**Seite 6: [3] Gelöscht**            **Christine Alewell**            **23.11.18 10:52:00**

**Seite 6: [3] Gelöscht**            **Christine Alewell**            **23.11.18 10:52:00**

**Seite 6: [3] Gelöscht**            **Christine Alewell**            **23.11.18 10:52:00**

**Seite 6: [4] Gelöscht**            **Microsoft Office-Benutzer**            **27.11.18 07:47:00**

**Seite 6: [4] Gelöscht**            **Microsoft Office-Benutzer**            **27.11.18 07:47:00**

**Seite 6: [5] Gelöscht**            **Christine Alewell**            **23.11.18 10:55:00**

**Seite 6: [5] Gelöscht**            **Christine Alewell**            **23.11.18 10:55:00**

| Seite 6: [6] Gelöscht | Christine Alewell | 23.11.18 10:58:00 |

| Seite 6: [6] Gelöscht | Christine Alewell | 23.11.18 10:58:00 |

| Seite 6: [7] Gelöscht | Christine Alewell | 23.11.18 10:58:00 |

| Seite 6: [7] Gelöscht | Christine Alewell | 23.11.18 10:58:00 |

| Seite 6: [8] Gelöscht | Microsoft Office-Benutzer | 26.11.18 09:53:00 |

| Seite 6: [8] Gelöscht | Microsoft Office-Benutzer | 26.11.18 09:53:00 |

| Seite 6: [9] Gelöscht | Christine Alewell | 23.11.18 11:01:00 |

| Seite 6: [9] Gelöscht | Christine Alewell | 23.11.18 11:01:00 |

| Seite 6: [9] Gelöscht | Christine Alewell | 23.11.18 11:01:00 |

| Seite 6: [9] Gelöscht | Christine Alewell | 23.11.18 11:01:00 |

| Seite 6: [9] Gelöscht | Christine Alewell | 23.11.18 11:01:00 |
|---|---|---|

| Seite 6: [9] Gelöscht | Christine Alewell | 23.11.18 11:01:00 |
|---|---|---|

| Seite 6: [9] Gelöscht | Christine Alewell | 23.11.18 11:01:00 |
|---|---|---|

| Seite 6: [9] Gelöscht | Christine Alewell | 23.11.18 11:01:00 |
|---|---|---|

| Seite 6: [10] Gelöscht | Microsoft Office-Benutzer | 15.11.18 09:17:00 |
|---|---|---|

| Seite 6: [10] Gelöscht | Microsoft Office-Benutzer | 15.11.18 09:17:00 |
|---|---|---|

| Seite 6: [11] Gelöscht | Christine Alewell | 23.11.18 11:05:00 |
|---|---|---|

| Seite 6: [11] Gelöscht | Christine Alewell | 23.11.18 11:05:00 |
|---|---|---|

| Seite 6: [12] Gelöscht | Microsoft Office-Benutzer | 20.11.18 13:12:00 |
|---|---|---|

| Seite 6: [12] Gelöscht | Microsoft Office-Benutzer | 20.11.18 13:12:00 |
|---|---|---|

| Seite 6: [12] Gelöscht | Microsoft Office-Benutzer | 20.11.18 13:12:00 |
|---|---|---|

| Seite 6: [12] Gelöscht | Microsoft Office-Benutzer | 20.11.18 13:12:00 |

| Seite 6: [12] Gelöscht | Microsoft Office-Benutzer | 20.11.18 13:12:00 |

| Seite 6: [13] Gelöscht | Microsoft Office-Benutzer | 20.11.18 13:15:00 |

| Seite 6: [13] Gelöscht | Microsoft Office-Benutzer | 20.11.18 13:15:00 |

| Seite 6: [14] Gelöscht | Christine Alewell | 23.11.18 11:18:00 |

| Seite 6: [14] Gelöscht | Christine Alewell | 23.11.18 11:18:00 |

| Seite 6: [15] Gelöscht | Axel | 19.11.18 11:50:00 |

| Seite 6: [15] Gelöscht | Axel | 19.11.18 11:50:00 |

| Seite 7: [16] Gelöscht | Axel | 26.11.18 06:43:00 |

| Seite 7: [17] Gelöscht | Microsoft Office-Benutzer | 20.11.18 13:24:00 |

| Seite 7: [18] Gelöscht | Christine Alewell | 23.11.18 11:26:00 |
|---|---|---|

| Seite 7: [19] Gelöscht | Christine Alewell | 23.11.18 11:30:00 |
|---|---|---|

| Seite 7: [20] Formatiert | Microsoft Office-Benutzer | 27.11.18 08:19:00 |
|---|---|---|

Schriftart: Kursiv

| Seite 7: [21] Formatiert | Microsoft Office-Benutzer | 27.11.18 08:19:00 |
|---|---|---|

Schriftart: Kursiv

| Seite 7: [22] Formatiert | Microsoft Office-Benutzer | 27.11.18 08:20:00 |
|---|---|---|

Schriftart: Kursiv

| Seite 7: [23] Formatiert | Microsoft Office-Benutzer | 28.11.18 09:56:00 |
|---|---|---|

Schriftart: Kursiv

| Seite 7: [24] Gelöscht | Microsoft Office-Benutzer | 28.11.18 09:57:00 |
|---|---|---|

| Seite 7: [25] Gelöscht | Microsoft Office-Benutzer | 28.11.18 09:59:00 |
|---|---|---|

| Seite 8: [26] Gelöscht | Christine Alewell | 23.11.18 11:44:00 |
|---|---|---|

| Seite 9: [27] Gelöscht | Microsoft Office-Benutzer | 22.11.18 13:21:00 |
|---|---|---|

| Seite 9: [28] Gelöscht | Axel | 26.11.18 07:51:00 |
|---|---|---|

---

## Author Response (AR2)

Comments to the Author:
Dear Dr. Birkholz,

We received a review of your revised manuscript. The referee was very pleased to find your nice improvements, but has still had severe concerns about the correction of Suess effect and a carbon exchange mechanism to explain the highly isotopically depleted long-chain fatty acid signatures in the deep sediments. So please read the following comments from the expert, and make a further revised manuscript with point-by-point responses.

Thank you again for your wonderful efforts to improve the manuscript.

Kind regards,

Koji Suzuki
Associate Editor
* * *
Dear editor, dear reviewer,

Thanks a lot for reviewing and editing our submitted manuscript. In the following we tried to split up the reviewer comments into suitable sections and did a point by point reply.
We think and hope that we now have found a good way to deal with the two main concerns of the reviewer, the missing source and the correction for the Suess effect. We completely removed the speculation about carbon exchange in fatty acid molecules in the sediment core and we also acknowledge the almost certain likelihood of an in-situ production of long chain FAs by bacteria, algae or other microorganisms in the lake. However, we rather would like to keep the hypothesis about the depleted FA origin from historical peatlands/riparian area in the manuscript and would appreciate, if you would agree on it. Regarding the Suess effect we have to admit, that our original calculation was wrong. We now have explained in more detail how our Suess effect estimation on soils was done and demonstrate why, from our perspective, the correction for the lake sediment samples is very difficult and can only be performed when the missing source and their isotopic signature would be known.
We hope that with this second revision we can convince you that the manuscript is suitable for being published in Biogeosciences.

Thank you again for investing so much time in improving our manuscript.

Kind regards,
Axel Birkholz, in the name of all Co-Authors.

Comments and over view:
The authors have generally responded well to the referee's comments and many of the concerns raised have been dealt with, appropriately. There is obviously a missing source in the polygons and that will affect the model output. It is Okay to state that there is an unknown source, but it would be better to complete the search and identify the most likely candidate. As I see it, the most likely candidate is the sediment bacteria that the authors speculated were exchanging the carbon isotopic signatures of the terrestrial FAs.

First of all, thanks a lot for the reviewer efforts to help making the manuscript better and more concise.
As shown in the 1st revised manuscript we hypothesized possible potential sources like algal production, bacterial production in the lake or production in historical wetlands/ reedbeds with isotopically depleted plant material (due to the use of depleted $CO_2$ (from bacterial methane oxidation) for photosynthesis).
For the new revised manuscript we removed the speculation about carbon exchange completely as we agree with the reviewer, that this is not based on scientific evidence.

*Deleted paragraph P10L36-P11L2*

Because of the fascination with trying to invoke this new source for isotopically depleted FAs in the

"The need to precisely identify sediment sources is especially important in eutrophic systems to enable efficient and targeted restoration measures." "While the eutrophication history of the Lake Baldegg has extensively been studied (), an in-depth confrontation of the lake evolution with the recent history of the catchment (including land-use and agricultural practices changes) has not yet been performed." "Our project aimed at filling these gaps. In this paper, the soil isotopic signatures of FAs characterizing the main land-uses of Lake Baldegg catchment are quantified and confronted to the evolution of the CSIA imprint of a 130-yrs long lake sediment sequence."
That fascination has led to speculation which is unsubstantiated but subsequently treated as fact.

No sorry, but we cannot completely agree with this. We can for a large part interpret the evolution of the CSSI signatures and the reflection of the land use change in this development and explain this in detail in the manuscript. Only part of the negative deviation is left unexplained and for this we try to develop explanatory hypotheses.

There are also errors in the calculation of the Suess effect.

We agree with you, but please see answers regarding the Suess effect below. where the reviewer is more detailed in his/her comment.

While the paper is important because of the in-depth study of the long-chain FAs in the sediment, the purpose of the paper has become buried in non-essential discussion and needs to be re-written in places to make it more concise and without speculation.

Specifics
The exchange concept has not been adequately dealt with. The authors are still speculating that a carbon exchange is the explanation for the highly isotopically depleted C28:0 fatty acid signature at depth in the sediment while ignoring scientific reasoning and weight of evidence that it is not. Because it is important to get this correct, I will explain. Look at the statement on Page 10; line 34 onwards:
"Here the example of van Bree et al. (2018) can be consulted, as they found a compound-specific enrichment of mostly C28:0 FA in the water column. Also the accompanying depletion of, in their case, C28:0 FA compared to the terrestrial C28:0 FA signal is hinting into the direction of an aquatic source. A diagenetic transformation of the FAs isotopic signal can also be speculated for the sediments older than 1940. Such an assumption would mean that these sediments would have been affected by carbon exchanges years to decades after their deposition, during the most severe eutrophic phases of the lake history. Why these exchanges would not have affected the younger sediments remains unexplained. And so far, no cases of such a diagenetic transformation have been described."
Statement: "C28:0 FA compared to the terrestrial C28:0 FA signal is hinting into the direction of an aquatic source". van Bree et al. (2018) are not just hinting at an aquatic source – they have documented it as an aquatic source and provided a strong "weight-of-evidence" case to support that conclusion.

In the new revised version, we will emphasize more clearly, that van Brees conclusions are also a likely scenario for Baldeggersee. We are not excluding or questioning van Brees theory. Please see in the last revised manuscript P7L18-36, where we already mentioned the same conditions in Lake Baldegg, like CO2 undersaturation and pH as in Lake Chala. And P8L1-10. Regarding the reviewer comment above, in our case it is mostly C26:0 and C24:0 FAs which are depleted. C28:0 is staying within the mixing polygon during that time and is only depleted during the isotopical excursion in the 1960s/70s.

*Changed "affected" to "masked" on P8L7.*
*And please see adapted chapter 4.2.*

Statement: "A diagenetic transformation of the FAs isotopic signal can also be speculated ". This speculation is unsubstantiated and incorrect. There are good reasons why "no cases of such a diagenetic transformation have been described" – because it doesn't happen. A diagenetic exchange of carbon atoms that changed the isotopic signature of the original molecule is impossible. If it could occur, the original C28:0 molecule would be broken into smaller chain-length pieces and no longer contribute to the C28:0 carbon pool.

Yes, we agree with the reviewer. As already stated above, we will remove this part completely.

You could, however, mask the original isotopic signature with a more depleted isotopic signature and the observed isotopic signature would appear to have changed if that more depleted isotopic source was not included in the modelling.

We agree with this point of view and changed accordingly chapter 4.2. and the conclusions.

Come back to first principles:
1) The FA exudate from plant roots labels the soil by binding on to it. The plant litter does not label the soil because the FAs in the litter are already incorporated into the plant matter and are not available as free acid radicles to bind to soil particles. Water sorting in lakes and bacterial decomposition processes remove the plant litter carbon rapidly but not the FA carbon bound to the soil.

We agree that FA exudate from plant roots label the soil. However above ground plant material (e.g., leaves, needle) can also end up as humus material transferred to the soil via degradation in the humus layer and then downward transport in the soil profile and eventual binding to soil particles of the mineral soil. However, in both cases, FAs bound to soil particles are useful labels of the terrestrial plant growing on the soil.

2) The isotopic signature of the FA in the sediment does not "change" going down the sediment core. Because the FA must be produced close to the sediment particle it is bound to, it is the source of the FA that is changing down the core. Hence, we can use CSIA to identify sources.

We totally agree with the statement, that the source is changing. If we have formulated imprecisely which caused misunderstandings in the text we are sorry and we have changed it accordingly.

3) The source of highly depleted d13C isotopic signatures in sediment is associated with methane production and the presence of methane oxidising bacteria (MOB) at the sediment-water interface. Under stratified water column conditions, the MOB can convert methane into CO2 (with the same highly depleted isotopic signature as the methane) at the oxic/anoxic interface in the water column. [Naeher et al. 2014: Tracing the methane cycle with lipid biomarkers in Lake Rotsee (Switzerland). Organic Geochemistry 66: 174–181] for use by algae such as cyanobacteria, which are known to produce C28:0 FAs, and other microbial processes, which produce long chain FAs – these will all have highly depleted isotopic signatures. Look at fig 10, fig 12 and fig 13 in van Bree et al. (2018). These figures show the timing of the production of highly isotopically depleted C26:0 and C28:0 to be after the collapse of an algal bloom in Autumn when their lake was thermally stratified and had probably achieved an anoxic (i.e., no oxygen but has oxygen containing molecules - SO4, NO3) hypolimnion. While van Bree et al. (2018) have linked the highly isotopically depleted long chain FAs to phytoplankton, other researchers have simply attributed the FA production to 'bacteria" [e.g., Petrišič et al. 2017: Lipid biomarkers and their stable carbon isotopes in oxic and anoxic sediments of Lake Bled (NW Slovenia). Geomicrobiology 34(7):1-12; Steger et al. 2015: Comparative study on bacterial carbon sources in lake sediments: the role of methanotrophy. Aquatic Microbial Ecology 76: 39–47; and many other papers]. Most researchers haven't looked at the longer chain fatty acids, which is why this paper is important.

Yes, thanks for these thoughts, we totally agree with you and thought we made these points, at least regarding van Bree et al. (2018) already in the revised manuscript. So far it was assumed, that the long chain FAs are mainly produced by higher plants (Eglinton and Eglinton, 2008) which is the reason why bacteria studies mainly looked after the shorter ones (cf. Naeher et al. 2014, Steger et al., 2015; literature the reviewer was providing). Thanks a lot for providing the publication of Petrisic et al. (2017), which we didn't know yet. We will incorporate this publication into our new revised manuscript as another case study which is providing more evidence for the ability of bacteria to produce long chain fatty acids in a lacustrine environment.

*Please see adapted chapter 4.2.*

4) The organisms with highly depleted long chain FAs produced in the lake during stratification will deposit on the sediment surface and be buried with the next input of sediment and detritus. That will lock the depleted isotopic signature in the sediment as a marker for that time/depth. As the water quality of the lake changes after the removal of the sewage input, the overlying bottom water became anaerobic (i.e., no oxygen or molecules with oxygen -e.g., SO4 and NO3) depriving

the MOB of the oxygen source needed to function. Consequently, the MOB probably became a minor contributor to the long-chain FA pool.
Statement: "Why these exchanges would not have affected the younger sediments remains unexplained". The above is the explanation - the younger sediment was not deposited when the lake was hypertrophic so there would be little if any affect from the MOB process.
Using this scenario, while sewage was entering Lake Baldegg, it is likely that the new source of long chain FAs, locally produced in association with MOB in the lake by the indirect utilization of isotopically highly depleted methane, has masked the stable isotope signatures of the original terrestrial long chain FAs.

Thanks a lot for this explanation. We agree, that it is very likely, that due to a change in the lakes trophic status, the presence or abundance of the organisms which are responsible for the depleted FAs has changed. We will incorporate that in the revised manuscript. However, the wastewater treatment plant was only introduced in the late 1960s.

*Please see P12L5-6 and revised Chapter 5 P13P25-P14L22*

It is likely that the isotopic signature from the MOB activity will remain relatively constant in the lake, while the sewage was being discharge into the lake. If this is correct, the MOB depleted isotopic signature would be the missing source end-member. It then follows that the isotopic mixing between allochthonous and autochthonous long chain FAs could result in the variability of the isotopic signatures observed in the sediment. The degree of variability being dependant on the amount of external inputs of long chain FAs added to the lake. As the water quality in the lake improves with remediation treatment, the methanogenic processes become smaller and the terrestrial signatures mask them as they become dominant.
To determine the amount of terrestrial source in the deep sediment, the most depleted isotopic signatures could be used as a proxy for the in-situ production by the MOB, as the missing source in the mixing model.

Using the most depleted signal as proxy for the MOB produced FAs is a point we cannot fully agree with. This would implement, that the MOB signal is not only masking the terrestrial signal but totally overprinting it. And this is not very likely, since the transport of terrestrial material into the lake, also if masked, will still be an important process. If we look in addition at reported d$^{13}$C values of methane influenced FAs produced by bacteria or algae (eg. Petrisic et al., 2017/van Bree et al., 2018) we would expect a more depleted signal. The possible range for the missing source is too big to be fixed to one point. Please see Fig S3.

This explanation is provided to help understanding of a set of very complicated limnological processes working in highly eutrophic lakes.

Recommendations:
1) The section speculating on carbon exchange should be rewritten without the speculation, or simply be deleted. Acceptance of the van Bree et al findings could be a way forward.

As stated above, we will remove this part and emphasize more clearly van Brees findings and interpretation.

*Deleted P10L36-P11L2*
*And we adapted chapter 4.2 and conclusions, chapter 5.*

2) The polygon test should be repeated with the isotopically depleted deep sediment signature as a proxy for the MOB end-member and the data re-modelled including that end-member.

As already pointed out above, we would rather assume a more depleted source, because from our perspective an exclusive source of bacterial FAs is not very likely. In Fig. S3 we show a possible range of the missing source. From our point of view a modelling of the FA contributions using the most depleted sediment values would be very speculative and therefore we refrain from doing the model in this manuscript.

Other specific points:
Page 3, Sentence starting on line 15 "Despite the …" is confusing and should be rewritten.
Suggested rewording "The introduction of an artificial oxygenation system into the lake water column in 1982 (Stadelmann et al., 2002) lead to the disappearance of the varves from 1995.

Despite a strong decrease of P concentrations in the lake to below 30 µg.l–1 as the result of lake external and internal measures, the lake has not yet fully recovered from eutrophication (Müller et al, 2014).

*Changed accordingly.*

Page 6, line 3: "mostly grasslands since years," do you mean "grasslands for many years"?

*Changed to "many years".*

Page 6, I am not sure why the authors persist with discussing the Suess effect. It is a side issue easily dealt with by applying the correction. It is distracting from the main theme of the paper. It appears as though this discussion is trying to justify the speculation of carbon exchange, which is wrong.

No, we for sure do not want to justify the carbon exchange. On the contrary, we want to continue using the CSIA method to track sediments. However, the Suess effect correction for soil derived compounds, transported to lake sediments is not straight forward and far from easily done. Suess effect correction to soil organic matter is based on the estimation of many (unknown) variables like e.g. soil incorporation rate of plant material and the actual effect of the Suess effect on the plants (cf Zhao et al., 2001, straw). Our concerns regarding this correction have nothing to do with speculations about carbon exchange (which we deleted already anyway). Please see below for our thoughts on the Suess effect correction of soil bound material.

The interpretation of the Suess effect, as presented at bottom of page 6 and top of page 7, is ambiguous. The purpose of the Suess correction in the CSIA studies is to normalise all the historical mixture data to the present day isotopic signatures so that present day land use library samples can be used to identify likely land use sources at different depths (i.e., time before present) in the core.
The cited paper (Zhao et al. 2001) correctly states the isotopic depletion since pre-industrial age (generally accepted as 1700 AD) has been 2.5‰. This is the same factor identified by Verburg (2007) and can be calculated by the polynomial equation in Verburg (2007). That equation describes an exponential curve with little change between 1700 and 1840 but faster change from then to the present. The Suess correction required for sediment collected in 1940 is -1.6 ‰ and -1.3‰ for 1965. Those corrections need to be added to those sediment data to enable them to be modelled using present day land use samples as the source library.
The statement on page 7, first paragraph "The resulting maximum effect is a depletion of 0.22‰ from 1840 until 2016 for the soils. For the lake sediments the maximum depletion is 0.16‰ between 1840 to 2010. The older the sediments the smaller is the Suess effect on them." These values are very different to those required to normalise the sediment mixture isotopic signatures to present day equivalents. This indicates a calculation error that needs to be remedied before any further modelling.
The paragraph continues "In 1965 the calculated Suess effect on the lake sediments is only 0.01‰"
By my calculations a correction of -1.3‰ is required. And more "If we assume the calculated effect on the soils to be uniform and take into account that it is smaller than the analytical uncertainty, and further is based on the estimation of parameters like soil incorporation rate and actual effect of the Suess effect on the plants, we think that in our case it can be neglected."
The Suess effect cannot be neglected. The calculations used by the authors to generate the Suess corrections are clearly wrong and need to be redone and then the modelling needs to be redone with the corrected data.

First we want to acknowledge that there was a misunderstanding on our side about how to apply correctly the Suess effect correction. Many thanks to the reviewer for his effort to make things clear.

In the new revised manuscript, we replaced the existing chapter about Suess effect with a discussion how to apply the Suess effect to the lake sediment core if (mainly) terrestrial compounds are traced in the isotopic study.

Soil derived fatty acids are produced as wax lipids or from roots and are incorporated into the soil. They carry the atmospheric signature of the time when they were produced. After their incorporation into the soil they contribute to the continuum of soil organic matter with FA, as a mixture of FAs with very different ages (e.g. older than 1000years and recently built ones). Turnover times of FAs in

soils might vary widely. In lacustrine sediments, and also in coastal sediments n-alkane/FA ages of more than 1000 years were found.

The residence time and the accompanying mixing of FA with different ages in the soil and the time lag during transport is dampening the Suess effect in the soils. Using the $d^{13}C$ signature of our soils in 2015 as anchor point and knowing the atmospheric $d^{13}CO_2$ curve (Feng 1998, Verburgh 2007) we estimated the actual "Suess effect" on the soils with 3 different turnover times 10, 30 and 100 years (Fig S4-7), which cover the range discussed in the literature (Lichtfouse, 1997; Six and Jastrow, 2002; Wiesenberg et al, 2004). We did not consider possible changes during photosynthesis due to the increase of $CO_2$ concentration in the atmosphere or changes in the soil organic carbon content (which might be assumed to be fairly stable in forest and grassland soils, but might, of course, vary in agricultural soils).

Following these assumptions, for the different turnover times, the theoretical isotopic change from 1840 until 2015, induced by the Suess effect, was calculated:
-0.7permil, -1.33permil, and -1.91permil respectively.

These calculations are necessary to apply any correction for the Suess effect on terrestrial compounds in lake sediments. Because every age model of a lake sediment core can only give us the time of deposition. For autochthonous produced material, the production date and sedimentation date is in most cases identical, therefore the (from the reviewer) suggested correction (Verburgh 2007) can be applied. However, the soil derived compounds like long chain n-alkanes and long-chain FAs, have experienced mixing and ageing in the soils before sedimentation and this has to be taken in account, as suggested above.

Like (McCaroll et. al., 2009) we removed the Suess effect from the soils and lake sediments (calculation back to 1840), instead of applying an artificial Suess effect to the sediments. We see the strongest effect on todays soils which are all moved to more positive values. The younger sediments experienced a similar, slightly smaller correction to more positive values. Older lake sediment samples before 1940 are less effected by the correction, due to lower rates of fossil fuel burning and thus a smaller Suess effect. Therefore for all different turnover rate scenarios we observe a more clear separation from the older lake sediment samples compared to the source soils. This would suggest an even stronger influence of in-situ produced FAs. But in general, the corrections for the Suess effect haven't changed the general pattern of the FA isotopic data.

To sum up, we think that this valuable discussion points even stronger to the conclusion, that at least before 1940, in-situ production of long-chain fatty acids occurred in considerable amounts in the Lake Baldegg.

Recommendation: The calculation errors must be fixed and the section should be re-written to explain clearly why the Suess correction is being used.
A suggestion for this text is: "The Suess effect alters the isotopic signature of the FAs in the sediment from 1840 by around 2.2‰, which is substantially less than the isotopic depletion measured in the deep sediment. The source of that isotopic depletion is unknown but is consistent with methanogenetic processes in the sediment and water column of hyper eutrophic lakes (van Bree et al. 2018)."

Alternatively, the whole section on the Suess effect could be deleted as unnecessary.

Please see our comment above.
There are a lot of uncertainties that come along with the simulation of the Suess effect on soils and sediments, e.g. different isotope fractionation due to higher $CO_2$ concentration (Zhao et al., 2001), different turnover times in the different source soils, assumptions about the stability of the soil organic carbon pool etc.
In addition, studies which tried to estimate the Suess effect on soils by measurements, (Bird et al, 2003; Wiesenberg et al., 2004) arrived only with a smaller isotopic effect, -0,2- -0,3permil on the organic material. And to our knowledge, so far no study using terrestrial biomarkers in lake sediments was applying such a correction.
In the case of both, aquatic and terrestrial produced compounds in lake sediments, a mixed approach to correct for the Suess effect would have to be applied. But all sources must be known and isotopically characterized.

Page 11, line 38: remove the reference to carbon exchange (sec.4.2.)
*Replaced "the carbon exchange" with "authochtonous production" on P11L36.*

Page 12, line 9: The microbial activity overprints . . . I agree that this is what is happening. A better word than "overprints" is "masks". Optional change, not essential but has clearer meaning for an English-speaking audience.

*Replaced "overprints" with "masks" on P12L7.*

Page 12, Sentences starting line 29: "Potentially the algae or other microorganism responsible for the production of the long-chain FAs was [were] not present anymore after 1940. Following the hypothesis with organic material originated from the riparian zone and/or wetlands, the explanation could be that with time, wetlands, evolved after the lake level changes drained themselves or were drained by the farmers until 1940 for the intensification of agricultural practices".
It is correct that the microorganisms responsible for the production of the isotopically depleted long-chain FAs were no longer present after 1940 because the hypolimnion had become anaerobic to a depth of 40 m, removing the supply of oxygen required for the MOB activity in the sediment. The hypothesis that the organic material originated from the riparian zone is unsubstantiated speculation. If it is feasible, it should be measured and the isotopic values used to support the speculation.

It is correct, that these riparian processes were not yet described in the literature. But the timing of the events, the selective depletion of first C24:0/C26:0 and then C28:0 later, leaves this hypothesis at least to be a possible one. Neither in the study of van Bree et al. (2018) nor in Petrisics et al. (2017) study the actual producers are clearly identified. We would therefore leave it in the manuscript and hope that you agree on that.

*We added more information on P7L11-13*

Page 13, paragraph starting line 13: This will need to be re-written along with other sections of the paper to ensure the conclusions and recommendations are appropriate. These should also relate back to the original objectives of the study. The title says "Plants or Bacteria?". This manuscript clearly shows that it is both Plants and Bacteria, and that is a good conclusion. Don't change the title.

We adapted the conclusions chapter and we emphasized in the new revised version, that it is most likely both, bacteria and plants which are contributing to the long-chain FA pool in the sediments, but want to leave the hypothesis with the riparian organic material in the manuscript. We related the conclusions back to the paper title, which we of course leave as it is, but emphasize that for a clear source attribution the isotopic signal of the missing source has to be known.

[revised manuscript text omitted]

---

## Author Response (AR3)

**Associate Editor Decision: Publish subject to minor revisions (review by editor)** (25 Mar 2019) by

Koji Suzuki

Comments to the Author:

Dear Dr. Birkholz,

The Referee #2 was very pleased to see your modifications, but this expert has still some concerns on your manuscript. So we would appreciate it very much if you could revise your manuscript following the suggestions from Referee #2 (see below), and send us the revised paper with your point-by-point responses.

Thank you again for your patience and nice efforts to improve the manuscript.

Kind regards,

Koji Suzuki

Associate Editor

Dear Prof Suzuki, dear reviewer,

Thanks a lot for this very positive response and for your further efforts to improve the present manuscript. We did our best to meet the reviewers concerns and hope and think we were successful with it. Please find below our point by point reply. But we wanted to emphasize here that we are glad, that the reviewer and us are now in complete agreement in the most important questions about the manuscript, namely that the Suess effect is not the cause for the isotopic deviation of the older sediments, and that there is an autochthonous source producing long chain FAs. Further we would like to point out that the whole section in the manuscript, discussing the Suess effect, arose from the reviewers request to correct the data for this effect, which is not a trivial task as we were trying to explain to the reviewer and to the public with the last revised manuscript. Thus, and because we believe that a lot of readers will wonder about the influence of the Suess effect on our data, we would like to leave this section in the new revised version of the manuscript. Lastly we wanted to emphasize, that we didn't do the modelling of the recent sediments due to the inability to use a model like MixSIAR, but we didn't want to overload the already very complex manuscript. In a next step we want to model the different contributions to the recent lake sediment but also for the contributing streams, for which we have a complete set of suspended sediments, sampled during every high flow event over the course of two years.

We hope, that we were able with this revision to bring the manuscript to a level where it can be published in Biogeosciences.

Kind regards,

Axel Birkholz in the name of all Co-authors.

---

Referee 2, revised manuscript 2018_288-version 3

Title: "Plants or bacteria? 130 years of mixed imprints in Lake Baldegg sediments (Switzerland), as revealed by compound-specific isotope analysis (CSIA) and biomarker analysis"

Authors: Marlène Lavrieux, Axel Birkholz, Katrin Meusburger, Guido L.B. Wiesenberg, Adrian Gilli,

Christian Stamm, Christine Alewell

*Comments and over view:*

*The authors have largely answered the referee's comments. There is obviously a missing source in the*

*polygons and this has been discussed with suggested sources, both allochthonous and autochthonous*

*being raised as possibilities. It is Okay to state that there is an unknown source.*

Thanks for these overall positive comments.

*The only remaining contentious issue is the question of the Suess effect, what it means to this study*

*and how it should be calculated.*

Please see below for our detailed answers.

*There is also some confusion in the authors understanding of how the CSIA data is incorporated into*

*the soil and thus how it is corrected.*

It seems we have differing understanding of how organic matter is incorporated into the soil and then bound to soil particles. Thanks for the interesting discussion and please see below for more detailed answers.

*Referring to section 4.2.3 The necessity of "Suess effect" correction for terrestrial lipids in lake*

*sediments"*

*Definitions: Terrestrial lipids are allochthonous, i.e., produced outside the lake. Autochthonous lipids*

*are those produced, by whatever mechanism, in the lake water column or lake sediment.*

*The lipids of concern are the polar fatty acids, which exude from plant roots and bind ionically to the*

*soil particles. They are sufficiently soluble that they can be moved by infiltration rainwater though the*

*surface soils until they bind to the soil particles. Once they attach to the soil particle, they cannot be*

*removed by natural processes, with the exception of bacterial consumption.*

We totally agree up to this point, may be with the slight exception that we would not exclude that organic matter bound to soil particles might be desorbed again, even from the relatively stable mineral clay-humus-complexes (please see works by Kalbitz et al, Kögel-Knaber et al., Guggenberger et al., Kaiser et al. etc.).

*Under those conditions the concentration of the FA will decrease BUT the isotopic signature of that FA*

*pool remains unchained.*

Yes, sure, we agree that concentrations of dissolved organic carbon (DOC) and thus also dissolved fatty acids decrease if organic matter is bound to soil particles and clay minerals and that at least according to today's knowledge no isotope fractionation is accompanied by this process.

*As a consequence of this binding, the FAs attached to the soil particles do not have a turnover time,*
*which would imply that they can be exchanged from the soil particles.*

Well, here we clearly have to disagree. Soil organic matter can sorb and desorb to soil particles, they
might even desorb from clay minerals, even though, of course the latter binding might be considered
pretty stable. Please see publication by Wiesenberg et al. (2004), where they determine turnover
times in soils by using d13C of the FAs. We were wondering if the term "turnover time" led to
misunderstandings, and we thus replaced it with "mean residence time" in the new revised
manuscript.

*The FAs are not waxes. Waxes, including n-alkanes, are non-polar and do not bind to the soil*
*particles. They are generally not soluble in water but form part of the soil humus.*

Well, macromolecular non-polar substances will be part of the colloids in soils and will thus be part of
the complex colloid dynamic in soils, soil solutions and waters. They might be hold back in soils, if we
have elluvial and illuvial processes, like e.g., podsolization or clay migration.

*When the soil is washed off the land, the soil particles labelled with the FAs are carried to the*
*downstream deposition zone, in this case, Lake Baldegg. During that transport mode, the light*
*humic/organic component is separated from the original bulk soil and will eventually settle on top of*
*the heavier soil particles in the lake.*

Well, we think that most of the time the binding between soil particles and FAs are pretty stable,
especially as fines are washed of preferentially. If generally or dominantly the FAs, would be
separated from the soil particles during detachment and transport, the whole concept of using them
as tracer would not work. And we do not think that there is any scientific evidence for this theory.

*Because that material is organic it will be decomposed rapidly by allochthonous bacteria. In contrast,*
*the FA labelled soil particles will remain isotopically unchanged although the concentration may*
*reduce over time as they are consumed by bacteria.*

Again, sorry, to disagree. As long as the FAs are bound to soil/ mineral particles, they will not be
degraded. However, IF they are desorbed and degraded, we have to assume that we will have at
least some slight isotopic fractionation (unpublished data from our group, will be submitted this
month, we are willing to share the data with the reviewer, if he/she is interested).

*The application of the Suess correction can only be applied to the allochthonous FAs, because they are*
*in contact with the atmosphere, which is experiencing the Suess effect. The autochthonous FAs may*
*experience some of the atmospheric Suess effects, but they will also experience the biogenically*
*altered CO2 from sediment decomposition processes and plant respiration.*

Well, as plants as well as algae respire the CO2 from the atmosphere, and most organic material in
the lake or in the soil has its origin from plant material, all organic molecules younger than 1840 will
be affected by the Suess effect. But we agree, that this potential Suess effect will then be overprinted
by the described possible effects in our lake system (P14L12-15).

*This effect has been correctly discussed by the authors. Since these latter processes have not been*
*assessed or documented, the authors have correctly not attempted to model the allochthonous*
*sources in the historical sediments.*

Thanks, we are glad you agree.

*(There is no reason why they couldn't model the contemporary sediment in the lake surficial*
*sediments, but they haven't.)*

Please see our detailed comments below, why a modelling of the recent sediments was not done
within this publication.

*With these basic facts as a starting point, the description of how the Suess effect affects the CSIA*
*values and how the correction for this effect should be calculated is misleading at best and is*
*misinformation, which should not be in this manuscript or any other.*
*Of particular concern is the lumping of the FA component bound to the soil particles with the organic*
*carbon from leaf litter and crop debris. When these are mixed in the soil they will have a turnover*
*time associated with the natural carbon cycle, as described by the authors. However, the authors*
*have combined the two distinctly different forms of soil carbon into a supposed soil label, and that is*
*wrong. Only the FAs bound to the soil particles act as labels. The leaf litter could have been blown into*
*the landscape from anywhere, thus contaminating the isotopic signature of the defined land use.*
*Most importantly, because this organic debris is non-polar, it cannot bind to the soil as a label, as*
*explained above.*
Sorry, but here we disagree. FA components bound to the soil particles originate from plants the
same way leaf litter or crop debris does. So all these substances will be prone to the Suess effect.

*Yes, the organic debris will breakdown and yes, there are FAs in the organic debris. The difference is*
*that those FAs are already ionically bound to other organic debris and are not able to bind to the soil*
*particles. If bacterial decomposition does release a free FA, it will be a shorter chain length than the*
*FA labels being measured and will not affect the CSIA values of the long chain length FAs being used*
*as labels.*
Yes, sure, we agree. This is why we only consider LCFA. Furthermore, from another study we were
conducting on intact leaves/needles, debris, humus and organic soil horizons we know that there are
plenty of free long chain fatty acids available in all fractions. This study will be submitted this April.

*The use of the Feng 1998 equation is not appropriate for the Suess correction. The equation published*
*by Verburg (2007) should be used. The use of the Verburg equation needs to be promoted as the*
*standard method for Suess correction rather than using older equations that do not have the level of*
*sophistication of the Verburg equation. The Verburg equation shows a present day isotopic depletion*
*of the CO2 in the atmosphere of just over 2.5‰ since the beginning of the industrial revolution, which*
*is generally accepted as 1700AD.*

It does not make a big difference, but we changed in the text accordingly and used Verburg (2007)
for correcting the data.

*The authors are correctly trying to eliminate the Suess effect as the cause of the much larger isotopic*
*depletion of some of the sediment FA tracers in the lake.*

We are glad that we agree with the reviewer in this really important point. And we think this is the
most important point within the whole discussion: that the Suess effect is not the cause of the
deviation in isotopic signatures of sources and sediments.

*This can be done with a simple statement just like that: e.g., "The isotopic depletion due to the Suess*
*effect is estimated to be about -2.5‰ since 1700AD (Verburg 2007) and is, therefore, unlikely to have*
*a substantial effect on the autochthonous production of FAs by lacustrine processes. Consequently,*
*the large isotopic depletion observed is most likely from an unknown autochthonous source or*
*associated with draining wetlands etc, etc . . ." Write it how you want. This is all that is needed in this*
*manuscript.*
*I strongly recommend that the misinformation in section 4.2.3 is removed by deleting everything from*
*page 9, line 23 starting "We thus want to discuss ".to." speculative" on page 10, line 28, and rewriting*
*the text in a form similar to that suggested above. Figure S4-7 and Table S4-5 may need to be revised*
*to match this textual change, if they are still needed.*
*The reason for the deletion is that the Suess effect can quickly be eliminated as a causal influence, as*
*suggested above, and is, therefore, not a significant component of the manuscript. As it is written, it*
*is a distraction and the reader would be justified in asking "Why did the authors put this in, it adds*
*nothing to the manuscript and is very confusing."*

Well, sorry, we feel that in combination with a correction for this effect, it is important to include the
discussion on the Suess effect, because even though the deviation we are observing is in the
opposite direction of the Suess effect and the Suess effect can thus not be an explanation at all, we
show in the graphs describing the Suess effect as a function of component mean residence time, that
the effect can have significant consequences and a correction for it should be discussed. Especially if
we have an aquatic source, which would make a valid correction almost impossible. And as far as we
know the Suess effect was never discussed for terrestrial biomarkers in lake sediments.

The whole section in the manuscript, discussing the Suess effect, arose from the Reviewers request
to correct the data for this effect, which is not a trivial task as we were trying to explain to the
reviewer and to the public with the last revised manuscript.

Already in the last revised manuscript we excluded the Suess effect as explanation for our negative
deviation with a simple statement. Please see P9L23-25 "Therefore, we would expect older lake
sediment samples to be relatively enriched (less depleted) in $\delta^{13}$C compared to our todays source
soils or sediments. As such, the Suess effect cannot explain our deviation of the isotopic signals
between source soils polygon and lake sediments."

*General comments:*
*Throughout the manuscript, the range between two values has been either a dash "-" or the word*
*"to". Because some of the numbers have negative values (e.g., -40.0 - -43.3‰), the word "to" should*
*be used consistently throughout the manuscript.*

we changed it accordingly.

*The abbreviation "FA" has been defined as "fatty acid" in the abstract only. This definition must be*
*repeated in the manuscript text at the first occurrence of FA. The abbreviation should then be used*

We changed it accordingly.

*The author's proof reading of their manuscript is not good. There are nine references which are not*
*cited in the text and two references that have no year. See "references section below".*
Extremely sorry for this. It happened due to the deletion of whole sections in this paragraph during
the revisions. We corrected for that in the revised manuscript.

*Specific points:*
*Page 6, Line 12: I disagree with the general statement that "The isotopic signature of the samples*
*older than 1940 and from 1964 - 1972 fall out of the source soils mixing polygon, making the use of a*
*mixing model to quantify the contribution of different land-uses to sediment inputs impossible.*
*Certainly, before 1972, source proportion modelling would be inappropriate until the missing source*
*has been found. However, where the more recent sediment CSIA data falls within the mixing polygon,*
*those data can be modelled, and should be for the completeness of this manuscript. I can recommend*
*the use of MixSIAR, without including concentration or any other "priors".*

You are totally right and we have to admit, that this might be one weak point of our manuscript.
However, we have a whole data set of river sediments from 5 rivers of the Baldegg Lake including
storm events and we would like to do the sediment source attribution with MixSIAR for the recent
lake sediments together with the river sediment data. To include all of this in the current manuscript
would certainly overload this current study and would make the manuscript much too long and
confusing. Even more so, as the uncertainty connected to the distribution of the source soil isotopic
signatures (e.g. not a clear spread polygon, but all signals on a more or less straight line), will need a
complex and in depth evaluation, interpretation and discussion.

*Page 8, Line 19: Why would you expect higher fatty acid concentrations in lake sediments compared*
*to source soils? Unless the autochthonous fatty acids are produced in the sediment, where they can*
*bind to the sediment particles, they will deposit on the sediment surface where they can be rapidly*
*destroyed by bacterial decomposition processes. This means that the concentrations in the lake will*
*decrease. Recommend that this sentence and the next are deleted as unnecessary and speculative.*

If there would be an aquatic source producing FAs, preferentially C26:0 and C24:0FA, and masking
the corresponding terrestrial FA signal in the lake sediment, we would expect higher concentrations
of these 2 FAs compared to C28:0FA (not absolute concentrations but relations between the
concentrations) which is obviously not as much affected as the two shorter ones. We will change the
wording in the manuscript for a better understanding (please see P8L18-21).

*Page 8, line 24: Change "-70+-15‰" to "-70 ±15‰"*
Changed accordingly.

*Page 8, lines 29 and 30 (and elsewhere): Where did the "n-fatty acids" expression come from. Please*
*remove the "n-" and just use "FAs" consistently throughout the manuscript.*

Changed accordingly.

*Page 13, Line 11: The sentence "As expanded above (Sect. 4.2.), a high discrepancy in isotopic values*
*between long-chain FAs of close chain-length points to a degradation of the isotopic signal."*
*No it doesn't point to degradation. It probably points to a variable amount of the unknown source or*
*other sources in the FA mixture in the sediment. The isotopic signal will only change through*
*fractionation, and fractionation will destroy the original long chain FAs producing shorter chain FAs,*
*which are not measured or included in the data.*

Yes, you are right, this was a relict of the original manuscript. Thanks a lot for pointing that out.

We changed accordingly P13L11-14

But there seems to be a major misunderstanding how isotope fractionation works. If
bacteria/enzymes preferentially use the lighter isotope, the remaining fraction of intact FAs will
increase in its stable isotope signal.

*Since part of section 4.2. should have been deleted, this sentence and the following sentences need to*
*be checked against the rewrite of section 4.2.3 for consistency. As written, this appears to be an*
*example of where earlier speculation and misinformation has been treated as fact, thereby*
*perpetuating misinterpretation of the otherwise good data.*

Please see above comment and changes.

*Page 9, Line 23: (Verburgh 2007) incorrect spelling of 'Verburg'*

We corrected accordingly.

*Page 13, line 29: The sentence "The CSIA signals of arable lands as well as orchards plot halfway*
*between grasslands and forests, which may render difficult to correctly attribute the sources of*
*sediment samples lying between grasslands and forests end-members."*
*They don't always plot in the same positions with the different isotopic tracers, therefore*
*discrimination should be possible. Have you tried to model the data using a stable isotopic mixing*
*model, such as "MixSIAR", just using the contemporary surface lake sediment as the lake*
*endmember? The surficial sediment will have received negligible effect from any possible*
*autochthonous source. Just looking at the isotopic signatures gives an indication of which sources are*
*present. Modelling the isotopic signatures gives a robust assessment of the isotopic proportions.*
*Using an isotope-to-soil proportion converter, as presented in Gibbs 2008, gives the proportional*
*contribution of each source soil to the sediment mixture. These calculations will be valid to a*
*date/depth of about 1970 and would add greatly to the usefulness of this manuscript.*

Please see line 211-220 why we would prefer not to do the sediment source attribution in this
manuscript. Of course, using a model like MixSIAR might provide you with results and probabilities,
but if the underlying data is not rigorously discriminating between the sources, the results of this
modelling exercise might be rather questionable. As such, this will for sure be a complex discussion and needs careful and in-depth evaluations and interpretation and is thus, in our opinion, beyond the
possibilities of this current manuscript. We think that a combination of the lake sediment data with
the river suspended sediments will yield better and more justified results.

*Page 14, line 3: The text "(2) would hint into the direction of an additional source with low C28:0 FA*
*concentrations . . .". The interpretation of the data doesn't "hint in the direction of" it 'indicates an*
*additional source'. That source doesn't have "low C28:0 FA concentrations" it has 'a depleted C28:0*
*FA isotopic signature'. Since concentration is not a factor when dealing with isotopic signatures, the*
*relationship between C28:0 and C24:0 and C26:0 FAs should be checked in the text statements and*
*discussion.*

If the additional source would have depleted isotopic d13C values of C28:0 FA, as the reviewer is
suggesting in his above comment, we would expect a deviation from the source soils as we do see for
C26:0 and C24:0FA. But, since the C28:0FA values all lie within the isotopic range of the sources we
conclude that the aquatic source isn't producing C28:0 FAs at all, or just in small amounts. We will
more clearly formulate it in the manuscript (please see P14L3-4).

*Page 14, line 8: "(iii) algae with depleted δ13C values due to described effects of hydoxilation reaction*
*of CO2 combined with high pH values in the epilimnion and CO2 undersaturation, . ."*
*This paragraph does not fit what you are describing and should be deleted.*
*Explanation: The high pH by itself does not alter the isotopic signature of C. High pH in the water*
*column is most likely due to consumption of dissolved CO2 by algae during photosynthesis, which is*
*what you are alluding to. The algal species that cause the highest pH are cyanobacteria, which are*
*bicarbonate adapted so that they can utilise bicarbonate (HCO3) when all the CO2 has been used.*
*Thus, they outcompete non-bicarbonate adapted algae such as greens and diatoms. However,*
*cyanobacteria are buoyant and tend to stay near the surface for high light and, since the majority of*
*the CO2 and HCO3 at the lake surface water will be of atmospheric origin, their isotopic signatures*
*will mostly reflect the atmospheric value -12‰ for fractionation. The cyanobacteria can come into*
*contact with isotopically depleted CO2 when the lake mixes in autumn releasing the nutrients*
*(especially P) and the methanogenically-derived isotopically depleted CO2 that have accumulated in*
*the bottom waters. (Aquatic macrophytes can also raise the pH if they are bicarbonate adapted).*

We are not sure if we understand the reviewers comment correctly. As described in van Bree et al.
(2018) (and Teranes et al. (1999) for oxygen in Lake Baldegg), during times of CO2 undersaturation
and high pH values in the lake water (as we find in Lake Baldegg) atmospheric CO2 might dissolve in
water and react with OH$^-$ to form HCO3$^-$ (Hydroxilation of CO2), which is strongly depleted in d13C  (-
15‰ compared to +8‰ during the reaction with H$_2$O). Aquatic microorganisms using this HCO3-
would be able to produce depleted FAs.

We will reformulate in the revised manuscript to avoid misunderstanding.

We rephrased P14L8-10 and P7L36-40.

*Page 14, line 14: The text "CSIA was proven to be not suitable to quantitatively unmix terrestrial*
*sources from the Lake Baldegg historic sediments, and thus to apportion the relative contribution of*
*different land-uses to the sedimentary archive as long as the isotopic signal of the missing source is*
*not known."*

*This is not strictly correct. It was not 'proven' that CSIA was not suitable to quantitatively unmix*

*terrestrial sources from Lake Baldegg historic sediment, the authors have stated that, because the*

*isotopic data from the historic sediments did not fall inside the mixing polygon, they would not model*

*them. That is a valid approach but does not constitute proof that CSIA was not suitable to*

*quantitatively unmix, it means that the data was unsuitable to be modelled using CSIA to*

*quantitatively unmix. Very important difference.*

You are right, we will re-formulate this sentence. Thanks for pointing this out.

We rephrased m P14L14-15.

*In this respect, there is no evidence in the manuscript that the authors have attempted to model any*

*isotopic data, including the data that did fall inside the polygon. This is a missing element that could*

*improve the paper.*

Please see above for our reasons, why we suggest not to include the modelling in this study.

*Notwithstanding this, the conclusion that the authors see the imprints of plants AND bacteria in the*

*Lake Baldegg sediments is an important finding and that conclusion can be drawn from the data*

*presented without modelling.*

Thanks.

*Reference not cited in text.* deleted

[revised manuscript text omitted]

---

## Author Response (AR4)

**Associate Editor Decision: Publish subject to technical corrections** (26 Apr 2019) by Koji Suzuki
Comments to the Author:
Dear Dr. Birkholz,

It is a great pleasure now to accept your manuscript for publication in Biogeosciences. Thank you very much for responding to the reviewers' comments so carefully and completely.

Before it is published, please consider my editorial comments below.

P1, L28 and hereafter: please add a comma immediately after e.g. (e.g., ...).

P1, L29: "Sphagnum" should be italic. Also "spec." should be replaced by "species".

P2, L7 and hereafter: please add a comma immediately after "i.e.".

P3, L21: Please add a period immediately after al (Müller et al., 2014).

P3, L27: Please add a space between 2 and m.

P19, L5: The "2" in "CO2" should be subscript.

P19, L5: The "13" should be superscript.

Thank you again for choosing the EGU-journal Biogeosciences.

Sincerely,

Koji Suzuki
Associate Editor

Dear Prof. Suzuzki,

thanks a lot for your very positive and encouraging response. I am very glad that we found a good solution for publishing the manuscript, which definitely gained quality during the review process.
In the following I address your final comments.

Thanks again for handling the case so seriously.
Kind regards,

Axel Birkholz
in the name of all Co-authors.

P1, L28 and hereafter: please add a comma immediately after e.g. (e.g., ...).
We changed accordingly.

P1, L29: "Sphagnum" should be italic. Also "spec." should be replaced by "species".
We changed accordingly.

P2, L7 and hereafter: please add a comma immediately after "i.e.".
We changed accordingly.

P3, L21: Please add a period immediately after al (Müller et al., 2014).
We changed accordingly.

P3, L27: Please add a space between 2 and m.
We changed accordingly.

P19, L5: The "2" in "CO2" should be subscript.
We changed accordingly.

P19, L5: The "13" should be superscript.
We changed accordingly.

[revised manuscript text omitted]